# Assessing Responses and Impacts of Solar climate intervention on the Earth system with stratospheric aerosol injection (ARISE-SAI): protocol and initial results from the first simulations

Jadwiga H. Richter[1], Daniele Visioni[2], Douglas G. MacMartin[2], David A. Bailey[1], Nan Rosenbloom[1], Brian Dobbins[1], Walker R. Lee[2], Mari Tye[1], Jean-Francois Lamarque[1]

[1] Climate and Global Dynamics Laboratory, National Center for Atmospheric Research, Boulder CO, USA

[2] Sibley School for Mechanical and Aerospace Engineering, Cornell University, Ithaca NY, USA

*Correspondence to:* Jadwiga H. Richter (jrichter@ucar.edu)

**Abstract.** Solar climate intervention using stratospheric aerosol injection is a proposed method of reducing global mean temperatures to reduce the worst consequences of climate change. A detailed assessment of responses and impacts of such an intervention is needed with multiple global models to support societal decisions regarding the use of these approaches to help address climate change. We present here a new modeling protocol aimed at simulating a plausible deployment of stratospheric aerosol injection and reproducibility of simulations using other Earth system models, Assessing Responses and Impacts of Solar climate intervention on the Earth system with stratospheric aerosol injection (ARISE-SAI). The protocol and simulations are aimed at enabling community assessment of responses of the Earth system to solar climate intervention. ARISE-SAI simulations are designed to be more policy relevant than existing large ensembles or multi-model simulation sets. We describe in detail the first set of ARISE-SAI simulations, ARISE-SAI-1.5, which utilize a moderate emissions scenario, introduce stratospheric aerosol injection at ~ 21.5 km in year 2035, and keep global mean surface air temperature near 1.5℃ above the pre-industrial value utilizing a feedback or control algorithm. We present here the detailed set-up, aerosol injection strategy, and preliminary climate analysis from a 10-member ensemble of these simulations carried out with the Community Earth System Model, version 2 with the Whole Atmosphere Community Climate Model version 6 as its atmospheric component.

## 1 Introduction

Solar climate intervention (SCI), or solar radiation modification, is a proposed strategy that could potentially reduce the adverse effects on weather and climate associated with climate change by increasing the reflection of sunlight by particles and clouds in the atmosphere. The recent National Academies of Sciences, Engineering and Medicine (NASEM) report on solar geoengineering research and governance (NASEM, 2021) calls for increased research to understand the benefits, risks and impacts of various SCI approaches. Stratospheric aerosol injection (SAI), which

aims to mimic the effects of volcanic eruptions on climate, has been shown to be a promising method of global climate
intervention in terms of restoring climate to present day conditions in global climate or Earth system models (e.g.:
Tilmes et al., 2018; MacMartin et al. 2019; Simpson et al., 2019). However, there still exist large uncertainties in
climate response and impacts (NASEM, 2021, Kravitz and MacMartin, 2020), and ensuing human and ecological
impacts (Carlson and Trisos, 2018). Due to the large internal variability of Earth's climate, the evaluation of SCI risks
and impacts requires large ensembles of simulations (Deser et al., 2012; Kay et al., 2015; Maher et al., 2021) and
Earth system models (ESMs) capable of simulating the key processes and interactions between multiple Earth system
components, including prognostic aerosols, interactive chemistry, and coupling between the atmosphere, land, ocean,
and sea ice. For studies of climate intervention using SAI, an accurate representation of the entire stratosphere,
including dynamics and chemistry, is needed to capture the transport of aerosols and their interactions with
stratospheric constituents such as water vapor and ozone (e.g.: Pitari et al., 2014).
The Geoengineering Model Intercomparison Project (GeoMIP) for many years has facilitated inter-model
comparisons of possible climate responses to SCI to examine where model responses to geoengineering were robust
and identify areas of large uncertainty. However, in order to ensure participation from multiple ESMs, the design of
GeoMIP simulations has often been simplified by utilizing solar constant reduction (Kravitz et al., 2013; Kravitz et
al., 2021) or prescription of an aerosol distribution (Tilmes et al., 2015) or a spatially uniform injection rate of $SO_2$
(i.e. continuous injection from 10ºN to 10ºS in the most recent G6sulfur experiments (Visioni et al., 2021b). Visioni
et al. (2021a) showed that solar dimming does not produce the same surface climate effects as simulating aerosols in
the stratosphere. Kravitz et al. (2019) showed that strategically injecting $SO_2$ at multiple locations to maintain more
than one climate target may reduce some of the projected side-effects by more evenly cooling at all latitudes; hence,
model experiments with plausible implementation of SCI are needed in order to assess risks and benefits of these
strategies.
The Geoengineering Large Ensemble (GLENS, Tilmes et al. 2018), which used version 1 of the Community
Earth System Model with the Whole Atmosphere Community Climate Model as its atmospheric component
(CESM1(WACCM), Mills et al. 2017), was the first large-ensemble (20-member) set of climate intervention
simulations carried out with a single ESM that interactively represented many of the key processes relevant to SAI
and has provided a community dataset for the examination of potential impact of SAI on mean climate and variability.
GLENS utilized sulfur dioxide ($SO_2$) injections that were strategically placed every year to keep the global mean
temperature, equator-to-pole, and pole-to-pole temperature gradients near 2020 levels in an effort to minimize the
surface temperature impacts of this intervention. However, GLENS has several experimental design issues that are
not aligned with realistic projections for Earth system outcomes that would provide more accurate representation of
possible real-world effects and impacts. Firstly, GLENS adopted a high emission scenario of RCP8.5 until 2100,
requiring a very large amount of stratospheric aerosols by the end of the century to offset the continuously increasing
emissions. Estimates for future emissions based on current commitments are lower than RCP8.5 (Hausfather and
Peters, 2020), and thus impact analyses, especially based on the last two decades of the GLENS, are likely to
overestimate the risks and adverse impacts of SAI. Additionally, in the GLENS simulations, intervention commenced
in 2020, adding another unrealistic element from a real-world standpoint. Furthermore, $SO_2$ injections were at 23-25
km altitude, which is technologically more difficult to achieve than a lower altitude injection (Bingaman et al. 2020).
Tilmes et al. (2020) has carried out simulations with $SO_2$ injections with CESM2(WACCM6) and GLENS-
like set-up for the Shared Socioeconomic Pathway SSP5-8.5 and SSP5-3.4-OS scenarios (O'Neill et al., 2016). Here
we propose a new SAI modeling protocol for a suite of simulations designed to simulate a more plausible
implementation scenario of SCI using SAI that can be replicated by other modeling centers. We denote the entire set
of current and future simulations conducted under this protocol as "Assessing Responses and Impacts of Solar climate
intervention on the Earth system," or "ARISE," with simulations of SAI denoted "ARISE-SAI". We anticipate that in
the future similar simulations utilizing other climate intervention methods such as Marine Cloud Brightening (MCB)
or Carbon Dioxide Removal (CDR), will result in ARISE-MCB or ARISE-CDR simulations respectively.  In addition,
we present preliminary results from the first set of these simulations carried out with the Community Earth System
Model, version 2 with the Whole Atmosphere Community Climate Model version 6 as its atmospheric component
(CESM2(WACCM6)).  The paper is structured as follows: section 2 provides an overview of ARISE-SAI protocol
including ARISE-SAI-1.5, section 3 describes the model used to describe the realization of ARISE-SAI-1.5 with
CESM2(WACCM6), section 4 shows surface temperature and precipitation in these simulations, and section 5 offers
a summary and conclusions.
**2 ARISE-SAI**
**2.1 Reference Simulations**
Evaluation of impacts of SCI requires a set of non-SCI reference simulations to enable comparison of impacts with
and without SAI. As motivated by MacMartin et al (2022), we use here the moderate Shared Socioeconomic Pathway
scenario of SSP2-4.5 for our simulations, which more closely captures current policy scenarios compared to higher
emission scenarios such as SSP5-8.5 (Burgess et al., 2020). SSP2-4.5, which marks a continuation of the
Representative Concentration Pathway 4.5 (RCP4.5) scenario, is a "middle-of-the-road," intermediate mitigation
scenario where "the world follows a path in which social, economic, and technological trends do not shift markedly
from historical patterns" (O'Neill et al., 2017), representing the medium range of future forcing pathways (O'Neill et
al., 2016).
**2.2 Protocol Overview**
The ARISE-SAI simulations are designed to simulate a plausible implementation scenario of SCI using SAI for
evaluation of potential climate intervention risks and impacts. MacMartin et al. (2022) described in detail the need for
various scenarios to evaluate impacts of SCI and five dimensions of SCI deployment options which include the
background climate-change scenario, desired target of cooling, start date of deployment, how cooling is achieved, and
other factors that could affect decisions. The proposed default ARISE-SAI protocols follow closely the recommended
scenario choices described in MacMartin et al. (2022) and describe details of implementation in Earth system models,
although different choices can be made in the future to expand the simulation set. In particular, the proposed ARISE-

SAI simulations utilize a moderate emission scenario, SSP2-4.5 (O'Neill et al., 2016) and cool the Earth to a global mean temperature target (TT) above preindustrial levels denoted in the specific name of the simulations (e.g.: ARISE-SAI-TT). For example, ARISE-SAI-1.5 and ARISE-SAI-1.0 simulations aim to maintain global surface temperatures at ~1.5℃ and ~1.0℃ above preindustrial levels respectively.

The protocol in the first ARISE-SAI simulations (without a delayed start) simulates deployment beginning in 2035 after the global surface temperature reaches ~1.5℃ above preindustrial levels,  the target proposed in the 2015 Paris agreement and described by the IPCC as an important threshold for climate safety (IPCC 2018). Simulations are carried out for 35 years (2035 - 2069), which is sufficient to consider both a transition period of ~10 years and a quasi-equilibrium of at least 20 years after the controller converges. Minimum recommended ensemble size is 3, although more members will allow for more thorough evaluation of impacts on variability.

## 2.3 ARISE-SAI-1.5

 The first ARISE-SAI simulations, ARISE-SAI-1.5 presented here, aim to keep the global mean temperature at ~1.5℃ above pre-industrial levels. There is uncertainty among Earth system models with regard to when Earth's global mean surface temperature ($T0$) will reach 1.5℃ above pre-industrial levels. The recent Intergovernmental Panel of Climate Change (IPCC) Sixth Assessment Report (AR6) (IPCC, 2021) finds that 1.5℃ over pre-industrial will very likely be exceeded in the near term (2021- 2040) under the very high greenhouse gas (GHG) emission scenario (SSP5-8.5) and likely to be exceeded under the intermediate and high GHG emissions scenarios (SSP2-4.5 and SSP3-7.0). The IPCC AR6 defines 1.5℃ as the time at which $T0$ will reach 0.65℃ above the historical reference period of 1995 - 2014. The $T0$ between 1995 - 2014 is 0.85℃ above the pre-industrial (PI) value defined as the 1850 - 1900 average in the observational record. Using 31 global models, Tebaldi et al. (2021) found that the average across models of when 1.5℃ will be reached is 2028 under the SSP2-4.5 scenario (using 1995-2014 as 0.84℃ rather than 0.85℃ above PI), but with considerable variation across models. To simplify future model intercomparisons, we choose the time period of 2020 - 2039 (or ~ 2030 levels) as our reference period of when $T0$ is ~ 1.5℃ above PI values and make that the target $T0$ in the ARISE-SAI-1.5 climate intervention simulations. We acknowledge that different climate models, with different baseline temperatures and rates of warming, might have different time periods in which they reach 1.5. Nonetheless, we recommend that the best way to achieve a meaningful and easy comparison between different models would be to use always their own model's 2020-2039 SSP2-4.5 period as a baseline over which to calculate the targets their ARISE-SAI-1.5 simulations. This way, the reference period is the same between models and the 2035 start date remains meaningful in every case.

In addition to keeping $T0$, the ARISE-SAI simulations aim to keep the north-south temperature gradient ($T1$), and equator-to-pole temperature gradient ($T2$) to those corresponding to the temperature target. This is achieved by utilizing a "controller" algorithm (MacMartin et al., 2014; Kravitz et al., 2017) that specifies the amount of $SO_2$ injection. This approach was used in GLENS and the simulations presented in Tilmes et al. (2020). The controller algorithm is freely available as described in the Code Availability section. Sulfur dioxide injections in the ARISE-SAI simulations are placed at four injection locations (15℃S, 15℃N, 30℃S, 30℃N) into one grid box at ~ 21.5 km altitude.

The injection latitudes are the same as used in GLENS and in previous studies examining the model's responses to
single-point $SO_2$ injections (Tilmes et al., 2017; Richter et al., 2017). These four injection locations are sufficient to
independently control the targets that we are trying to achieve (Kravitz et al., 2017). These four injection locations
have also been demonstrated to be sufficient to produce the optical depth patterns that independently control the targets
that we are trying to achieve in various versions of CESM(WACCM) (MacMartin et al., 2017; Zhang et al., 2022;
MacMartin et al., 2022). The prescribed injection altitude is estimated to be achievable by existing aircraft
technologies that could be adapted for climate intervention use (Bingaman et al., 2020). After each year of simulation,
the algorithm calculates the global mean temperature, T0, north-south temperature gradient, T1, and equator-to-pole
temperature gradient, T2, and based on the deviation from the goal, specifies the annual values of injections at the
four locations for the subsequent year. T1 and T2 were defined in Kravitz et al. (2017), Equation 1.

**2.4 Recommended Output**
Comprehensive monthly output as well as high-frequency output for analysis of high-impact events (described in
detail in the Data Records section) is needed for analysis of SCI impacts on the Earth System. Acknowledging
limitations of various modeling centers, we recommended a minimum set of monthly-mean output fields in Table A1
in the Data Records section and include the full comprehensive output list that was created with the CESM2(WACCM)
simulations based on input from the broader community. All model output for the simulations should be provided in
NetCDF format. All variables should be in time-series format, with one variable per file. 3-dimensional atmospheric
output should be on the original model levels or on standard CMIP6 levels. For monthly atmospheric output,
information on aerosol microphysics (which is not a standard CMIP6 output) is also very relevant for diagnostics of
the aerosols' behavior under SAI; for instance, CESM2(WACCM6) includes as standard output the mass and number
concentration for all aerosol modes and the aerosol effective radius. Other modeling centers should consider providing
this (model specific) information as well. In addition, higher-frequency (daily averaged, 3-hourly averaged, 3-hourly
instantaneous, and 1-hourly mean) output is desired for the atmospheric model that will enable analysis of extreme
events (e.g.: Tye et al. 2022). The atmospheric output at various time frequencies is described in Appendix A, Tables
A2 - A5. Daily averaged output of land model variables is shown in Tables A6 and A7, whereas 6-hourly output from
the land model is listed in Table A8. Tables A9 and A10 show the daily output from the ocean and sea-ice models
respectively. The table captions describe which output is specific to ARISE-SAI-1.5 and the new five SSP2-4.5
CESM2(WACCM6) ensemble members, and which is common to all simulations. An online table showing all the
output fields for the simulations, along with their description and units, is at:
https://www.cgd.ucar.edu/ccr/strandwg/WACCM6-TSMLT-SSP245/.
**2.5 Additional ARISE-SAI simulations**
The ARISE-SAI-1.5 simulations described above are likely to be most relevant to policy makers and hence
reproduction of the experiments in multiple models is desired. ARISE-SAI simulations are already being performed
with the UKESM model. ARISE-SAI-1.0 simulations as well as ARISE-SAI-1.5-2045, with start of intervention
delayed by 10 years, are in progress with CESM2(WACCM). A subset of simulations describing these different initial
conditions and targets is discussed in MacMartin et al. (2022) using a slightly more simplified version of
CESM2(WACCM6).

**3. ARISE-SAI-1.5 with CESM2(WACCM6)**
We present here the details of implementation of ARISE-SAI-1.5 simulations in CESM2(WACCM6).

**3.1 Model Description**
CESM2(WACCM6) is the most comprehensive version of the NCAR whole atmosphere ESM and is described in
detail in Gettelman et al., 2019; Danabasoglu et al., 2020. CESM2(WACCM6) was used to contribute climate change
projection simulations to the Coupled Model Intercomparison Project Phase 6 (CMIP6) (Eyring et al., 2016).
CESM2(WACCM6). CESM2(WACCM6) is a fully coupled ESM with prognostic atmosphere, land, ocean, sea-ice,
land-ice, river and wave components. The atmospheric model, WACCM6, uses a finite volume dynamical core with
horizontal resolution of $1.25^o$ longitude by $0.9^o$ latitude. WACCM6 includes 70 vertical levels with a model top at 4.5
$\times 10^6$ hPa (~ 140 km). Tropospheric physics in WACCM6 are the same as in the lower top configuration, the
Community Atmosphere Model version 6 (CAM6). CESM2(WACCM6) includes a parameterization of non-
orographic waves which follows Richter et al. (2010) with changes to tunable parameters described in Gettleman et
al. (2019). Parameterized gravity waves are a substantial driver of the quasi-biennial oscillation (QBO) which is
internally-generated in CESM2(WACCM6). CESM2(WACCM6) includes prognostic aerosols which are represented
using the Modal Aerosol Model version 4 (MAM4) as described in Liu et al. (2016). This includes four modes, of
which only three are used for sulfate: Aitken, Accumulation and Coarse mode. In the stratosphere, CESM(WACCM6)
includes a comprehensive interactive sulfur cycle, as described for instance in Mills et al. (2016); this allows for $SO_2$
oxidation (with interactive OH concentration) and subsequent nucleation and coagulation of $H_2SO_4$ into sulfate aerosol
(allowing for inter-mode transfer), which are then removed from the stratosphere through gravitational settling and
large-scale circulation. A more indepth analysis of the size distribution and vertical distribution of sulfate aerosols
under $SO_2$ injections has been performed in Visioni et al. (2022) (for single-point injections at the same latitudes and
altitudes as those described in these simulations), also compared with results from other models with similar aerosol
microphysics (UKESM1 and GISS), highlighting that in CESM2(WACCM6) the produced stratospheric aerosol are
mainly found in the Coarse mode. CESM2(WACCM6) also includes a comprehensive chemistry module with
interactive tropospheric, stratospheric, mesospheric and lower thermospheric chemistry (TSMLT) with 228 prognostic
chemical species, described in detail in Gettleman et al. (2019).
The ocean model in CESM2(WACCM6) is based on the Parallel Ocean Program version 2 (POP2; Smith et
al., 2010; Danabasoglu et al., 2012; Danabasoglu et al., 2020). The horizontal resolution of POP2 is uniform in the
zonal direction (1.125°) and varies from 0.64° (occurring in the Northern Hemisphere) to 0.27° at the Equator. The
ocean biogeochemistry is represented using the Marine Biogeochemistry Library (MARBL), which is an updated
implementation of the Biochemistry Elemental Cycle (Moore et al., 2002; 2004; 2013). CESM2 uses version 3.14 of
the NOAA WaveWatch-III ocean surface wave prediction model (Tolman, 2009). Sea-ice in CESM2(WACCM6) is
represented using CICE version 5.1.2 (CICE5; Hunke et al., 2015) and uses the same horizontal grid as POP2.
CESM2(WACCM6) uses the Community Land Model version 5 (CLM5) (Lawrence et al., 2019). CLM5
includes a global crop model that treats planting, harvest, grain fill, and grain yields for six crop types (Levis et al.,
2018), a new fire model (Li et al., 2013; Li and Lawrence, 2017), multiple urban classes and an updated urban energy
model (Oleson & Feddema, 2019), and improved representation of plant dynamics. The river transport model used is
the Model for Scale Adaptive River Transport (MOSART; H. Y. Li et al., 2013).
**3.2 Reference simulations**
A 5-member reference ensemble with CESM2(WACCM6) and the SSP2-4.5 scenario was carried out as part of the
CMIP6 project for years 2015 - 2100. Surface temperature evolution and equilibrium climate sensitivity in these
simulations are described in detail in Meehl et al. (2020). We carried out an additional 5-member ensemble of these
simulations from years 2015 - 2069 with augmented high-frequency output for high-impact event analysis, as well as
additional output for the land model to match the SCI simulations. The additional 5-member ensemble was branched
from the three existing historical CESM2(WACCM6) simulations in the same manner as the first 5-member ensemble,
but with an addition of small temperature perturbations for each ensemble member ([6, 7, 8, 9, 10] $\times 10^{-14}$ K,
respectively), at the first model timestep. CESM2 ranks highly against other CMIP6 models in the ability to represent
large scale circulations and key features of tropospheric climate over the historical time period (e.g.: Simpson et al.,
2020; Duviver et al., 2020; Coburn and Pryor 2021).
**3.3 ARISE-SAI-1.5 Simulations**
In CESM2(WACCM6) $SO_2$ injections were placed at 180$^o$ longitude and bounded by two pressure interfaces: 47.1
hPa and 39.3 hPa (approximate geometric altitude at gridbox midpoint of 21.6 km). Based on the 2020 - 2039 mean
of the SSP2-4.5 simulations with CESM2(WACCM6), the surface temperature targets for the ARISE-SAI-1.5
ensemble for T0, T1, and T2 are 288.64 K, 0.8767 K, and -5.89 K, respectively. As noted in section 2.3, we recommend
that T0, T1, and T2 targets for other models reproducing ARISE-SAI-1.5 simulations are based on the 2020 – 2039
average from their SSP2-4.5 simulations.
The first five members of ARISE-SAI-1.5 simulations were initialized in 2035 from the first five members
(001 to 005) of the SSP2-4.5 simulations carried out with CESM2(WACCM6); hence, all had different initial ocean,
sea-ice, land, and atmospheric initial conditions on January 1, 2035. Similarly to the SSP2-4.5 simulations, subsequent
ensemble members (006 through 010) were initialized from the same initial conditions as members 001 through 005,
respectively, with an addition of a small temperature perturbation to the atmospheric initial condition to create
ensemble spread.

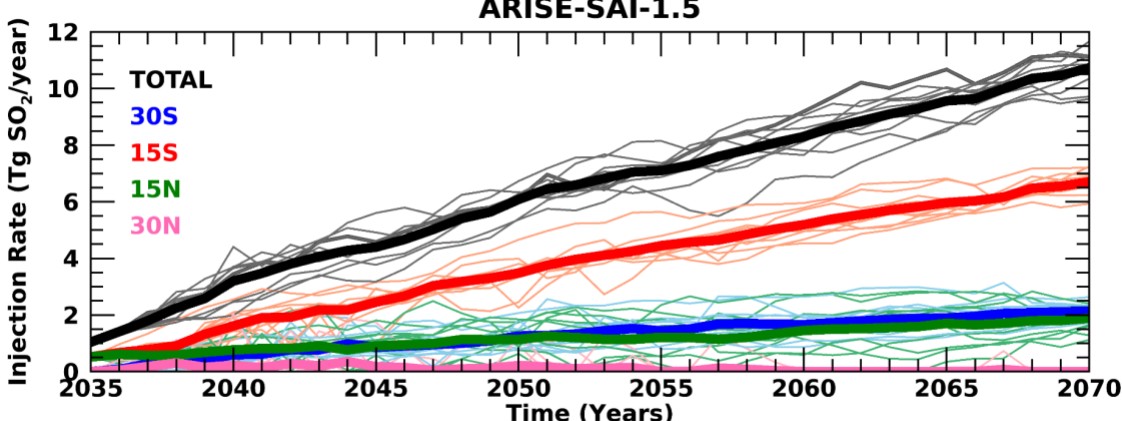

**Figure 1:** SO$_2$ injection rate as a function of time in ARISE-SAI-1.5 simulations at 30°S (blue), 15°S (red), 15°N (green), 30°N (pink), and total (black). Thin lighter colored lines represent individual ensemble members, whereas thick lines show the 10-member ensemble mean.

The amount of SO$_2$ injection in the ARISE-SAI-1.5 simulations chosen by the controller algorithm is shown in Figure 1. The majority of SO$_2$ is injected at 15°S, with an approximate linear increase from 0.5 Tg SO$_2$ per year in 2035 to 6 Tg SO$_2$ per year in 2069. SO$_2$ injections at 30°S and 15°N are about ⅓ of that injected at 15°S. Throughout all the ARISE-SAI-1.5 simulations, the amount of SO$_2$ injection at 30°N is very small, less than 0.5 Tg SO$_2$ per year, diminishing to nearly zero by the end of the simulations. The distribution of SO$_2$ across the four injection latitudes in ARISE-SAI-1.5 is very different from that in GLENS (Tilmes et al., 2018) despite having the same goals for the controller. In GLENS, the majority of SO$_2$ was injected at 30°S and 30°N, with a significant amount at 15°N, and almost none at 15°S; that is, GLENS required more injection in the Northern Hemisphere than the Southern in order to maintain the interhemispheric temperature gradient T1, whereas ARISE-SAI-1.5 requires more injection in the Southern Hemisphere to maintain T1. GLENS also required more SO$_2$ injection at 30°N/30°S to maintain T2 than is required in ARISE-SAI-1.5. It is unclear at this time how much of this difference is a result of the different model version and how much is a result of changes in the forcing between RCP8.5 and SSP2-4.5.

**4 Initial Results**

One of the intents of ARISE-SAI simulations is to provide the broader community a data set for examining various impacts of SCI on the multiple components of the Earth system. Below we present basic diagnostics that verify that the SO$_2$ injections and controller are working as intended, and we describe how well the temperature targets are being met in CESM2(WACCM6). Detailed analysis of the simulations is left for future work.

**4.1 Stratospheric Aerosols**


Injection of sulfur dioxide into the stratosphere results in the formation of sulfate aerosols, which are transported by
the stratospheric Brewer-Dobson circulation (Andrews et al., 1987; Tilmes et al., 2017). The dominance of $SO_2$
injections at 15ºS in ARISE-SAI-1.5 results in a stratospheric sulfate ($SO_4$) increase that primarily occurs in the
southern hemisphere, with the majority of $SO_4$ concentrated near the primary injection location (Figure 2a, 2b).
Averaged over the 2035 - 2054 period, there is a peak $SO_4$ increase of 25 mg-S/kg air (Fig 2a) relative to the 2020 -
2039 mean, and averaged over 2050 - 2069 an $SO_4$ increase of 48 mg-S/kg air is found near 15ºS, 40 hPa (Fig 2b).
The zonally averaged latitudinal distribution of the increase in the column of $SO_4$ is shown in Figures 2c, d; both
figures show the strong hemispheric asymmetry, and also a double peak at around 15ºS and one near 50ºS. The peak
near 15ºS is due to the predominant location of the injection, and matches the peak in concentration, the latter is due
to the largest vertical stratospheric layer over which $SO_4$ is spread out (between 10 and 22 km) compared to the layer
in the tropical stratosphere (between 18 and 26 km). Integrated over 20-year periods of ARISE-SAI-1.5 simulations,
there is little difference in the latitudinal distribution of column $SO_4$ between the various ensemble members, but
amplitude differences of up to 15% exist (not shown), reflecting variability in the amount of $SO_2$ injection at each
location and small differences in the stratospheric circulation.

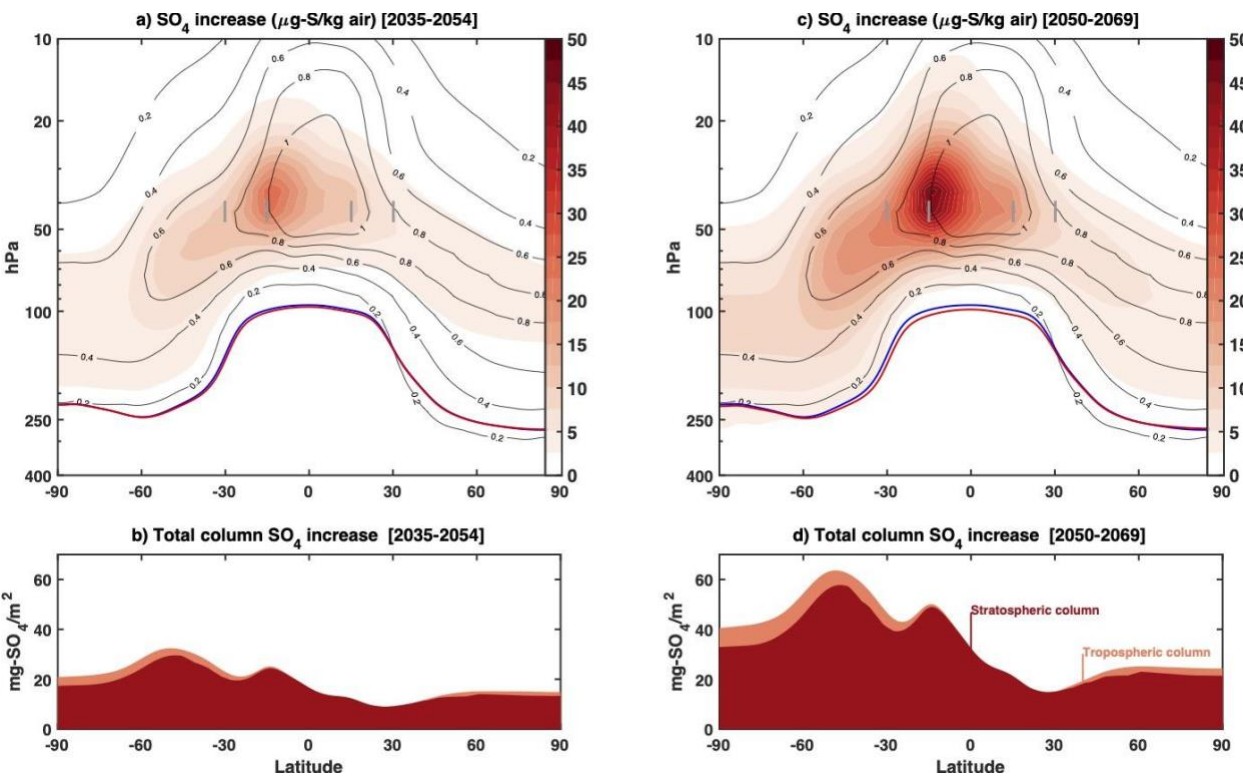


**Figure 2:** Zonal mean stratospheric $SO_4$ concentration increase (in μg-S/kg of air) in (a) 2035-2054 and (c) 2050-
2069 relative to the 2020 - 2039 mean. Black contour lines show the background concentration in 2020-2039. Blue
line shows the annual mean tropopause height in the control period; the red line shows the annual mean tropopause
height in the ARISE simulation in 2035-2054 and 2050-2069, respectively. Gray shadings indicate the grid-boxes
where $SO_2$ is injected. Zonal mean total increase in the column burden of sulfate (in mg-$SO_4$/m²) for (b) 2035 - 2054
and (d) 2050 - 2069. The contribution to the column increase is shown in dark red, for the fraction located in the
stratosphere, and in orange for the fraction located in the troposphere.
**4.2 Meeting temperature targets**
Global mean surface temperature, the inter-hemispheric temperature gradient, and equator-to-pole temperature
gradients for the SSP2-4.5 and ARISE-SAI-1.5 simulations are shown in Figure 3. There is a notable difference in
behavior of T1 and T2 in the SSP2-4.5 simulations as compared to the RCP8.5 simulations with CESM1(WACCM)
(not shown). In the CESM1(WACCM) simulations with RCP8.5, T1 and T2 were increasing steadily with time of
simulation, reaching a change in T1 of nearly 0.45 K, and a T2 change of 0.3 K by 2070 relative to ~ 2020 - 2039
mean (Tilmes et al. 2018). In contrast, T1 and T2 in the SSP2-4.5 simulation are increasing much more slowly, less
than 0.05 K for T1 and less than 0.1 K for T2 between the reference period (2020-2039) and 2070. The more moderate
(SSP2-4.5) emission scenario used in the CESM2(WACCM6) control simulations partially explains the slower
increase of T1 and T2 with time, however not all. Simulations with CESM2(WACCM6) and SSP5-8.5 scenarios also
show a much slower increase of T1 and T2 as compared to CESM1(WACCM) with RCP8.5. Differing modeling
physics, in particular cloud feedbacks, between CESM1 and CESM2 are key differences that could lead to the
differences in projected spatial patterns of surface warming between the two model configurations, as well as changes
in the Atlantic Meridional Overturning Circulation as discussed in Tilmes et al. (2020). Additional simulations with
CESM2 and RCP emissions have been performed to understand the relative role of differences in forcing and
differences in model physics on projected spatial patterns of global mean temperature and other variables between
CESM1 and CESM2. A detailed discussion of the reasons behind the model dependence in injection strategy in
GLENS, CESM1(WACCM) and ARISE-SAI-1.5, CESM2(WACCM6) simulations can be found in Fasullo and
Richter (2022). They show that the main contributors to the differences are: rapid adjustment of clouds and rainfall to
elevated levels of carbon dioxide, dynamical responses in the Atlantic Meridional Overturning Circulation (AMOC)
and differences in future climate forcing scenarios.

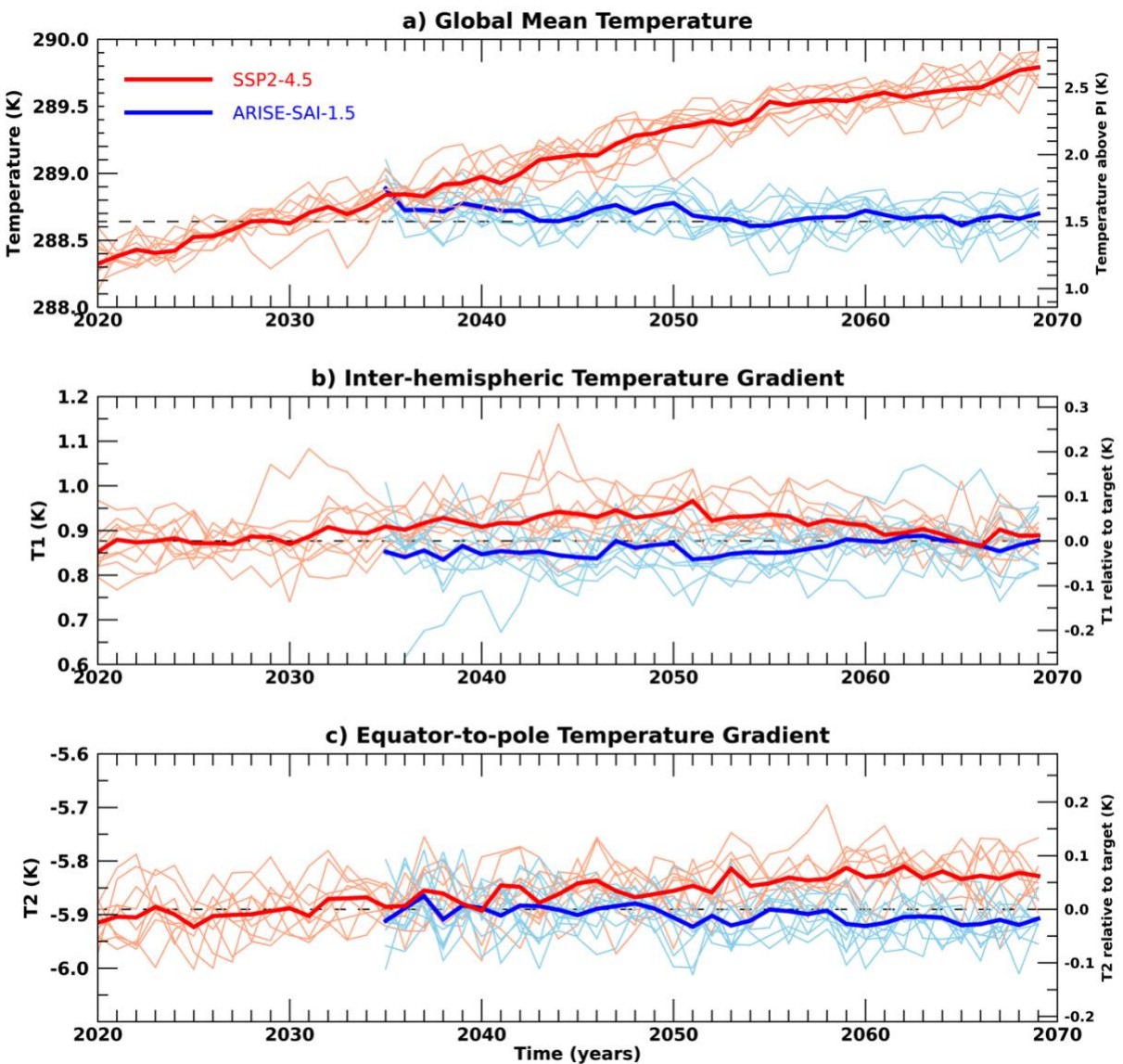


**Figure 3:** Global mean a) surface temperature, b) inter-hemispheric temperature gradient, T1, and c) equator-to-pole

temperature gradient, T2, for SSP2-4.5 (red) and ARISE-SAI-1.5 (blue) simulations. Thin lines represent individual

ensemble members, whereas the thick lines show the ensemble mean.


The differences between the projected surface temperature patterns in CESM2 as compared to CESM1 have

implications for climate intervention. Since the changes in T1 and T2 targets differ between the CESM1(WACCM)

and CESM2(WACCM6) future simulations, the controller selects different $SO_2$ injection locations to best counteract

these changes. Injections needed to offset increasing T1 and T2 in CESM1(WACCM) required primarily injections at

30°S and 30°N, whereas a small change in T1 and T2 relative to the 2020 - 2039 period in CESM2(WACCM6), SSP2-

4.5 requires injections primarily at 30°S. The $SO_2$ injections applied in ARISE-SAI-1.5 do a very good job at keeping

the global mean temperature, T1 and T2 at the target levels. This is demonstrated by the blue lines in Figure 2. There

is a fair amount of variability among the individual ensemble members (thin light blue lines) in their ability to meet
the global mean, T1 and T2 targets, however the ensemble mean (thick blue line) shows very good agreement between
these variables and their target values.

**4.3 Surface temperature and precipitation**

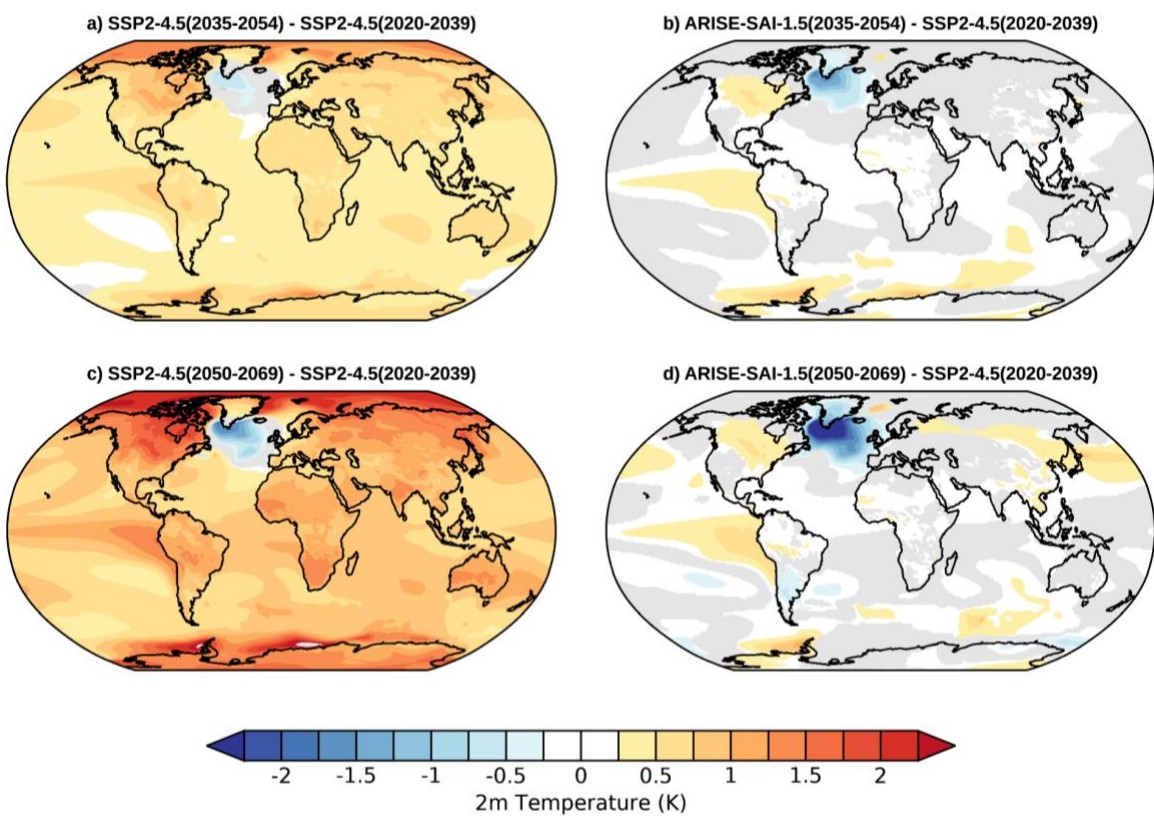


**Figure 4:** Ensemble and annual mean surface (2m) temperature differences between a) SSP2-4.5 (2035-2054) and
SSP2-4.5 (2020-2039), b) ARISE-SAI-1.5 (2035-2054) and SSP2-4.5 (2020-2039), c) SSP2-4.5 (2050-2069) and
SSP2-4.5 (2020-2039), and d) ARISE-SAI-1.5 (2050-2069) and SSP2-4.5 (2020-2039). Gray shading indicates
regions where the differences are not statistically significant at the 95% level using a two-sided Student's t test.

Figure 4 shows the ensemble and annual mean surface temperature changes for two time periods, 2035 - 2054 and
2050 - 2069, during the SSP2-4.5 and ARISE-SAI-1.5 simulations relative to the 2020 - 2039 period. Fig 4 a, c show
the steady increase in surface temperature with time over the majority of the globe, with the largest warming occurring
in the Northern Hemisphere high latitudes. The North Atlantic is the only region of the globe that is cooling in the
21st century. This "warming hole" in the North Atlantic is a feature of several of the recent generation Earth system
models and is attributed to the AMOC (Drijfhout et al. 2012, Chemke et al. 2020, Keil et al. 2020). Specifically, in a
warming climate with a reduction in the deep water formation, the AMOC weakens. This results in less heat transport
into the Northern North Atlantic, producing cooler temperatures that oppose the anticipated effects of global warming.
Figures 4b and 4d demonstrate the success of the SAI strategy in keeping the global temperatures near the 2020 - 2039
average, or at ~ 1.5 K above pre-industrial values. In ARISE-SAI-1.5, near surface annual mean temperature
throughout the entire simulation is within 0.5 K of that goal over the majority of the globe. The largest exception to
that is the North Atlantic warming hole, where surface temperatures remain cooler relative to the northern North
Atlantic than in the present day; while AMOC strength is partially recovered under SAI relative to SSP2-4.5, it is not
fully restored back to present-day conditions. In addition, in the ensemble mean, ARISE-SAI-1.5 simulations show
residual warming over North America, as well as over Eastern South Pacific Ocean (off the coast of South America),
and in parts of Antarctica as compared to the 2020 - 2039 period. Residual changes relative to the target period from
the application of SAI are expected, as SAI can not perfectly reverse the effects of increasing greenhouse gases.

The precipitation changes in SSP2-4.5 and ARISE-SAI-1.5 simulations for the same time periods examined

for surface temperature changes are shown in Figures 5 and 6. Consistent with prior similar studies, SSP2-4.5
simulations show primarily an increase of precipitation in a warming climate, with the largest increases along the
Equatorial Pacific Ocean, and a strong drying region northward of that (Figs 5, 6a,c). In ARISE-SAI-1.5, consistent
with previous studies (Kravitz et al., 2017; Lee et al. 2020), restoring global mean temperature is associated with an
overall decrease in annual mean precipitation (Fig 5), however regionally both increases and decreases occur. In
ARISE-SAI-1.5, the increased precipitation across the Equatorial Pacific seen in SSP2-4.5 decreases in magnitude,
but is still a persistent feature. ARISE-SAI-1.5 also shows drying north and south of that region as well as intensified
drying over Northern South America, South Africa, Indian Ocean south of the Equator and northernmost Australia.
The Indian Ocean north of the Equator and India are projected to be wetter in ARISE-SAI-1.5 as compared to the
2020 - 2039 period of SSP2-4.5.

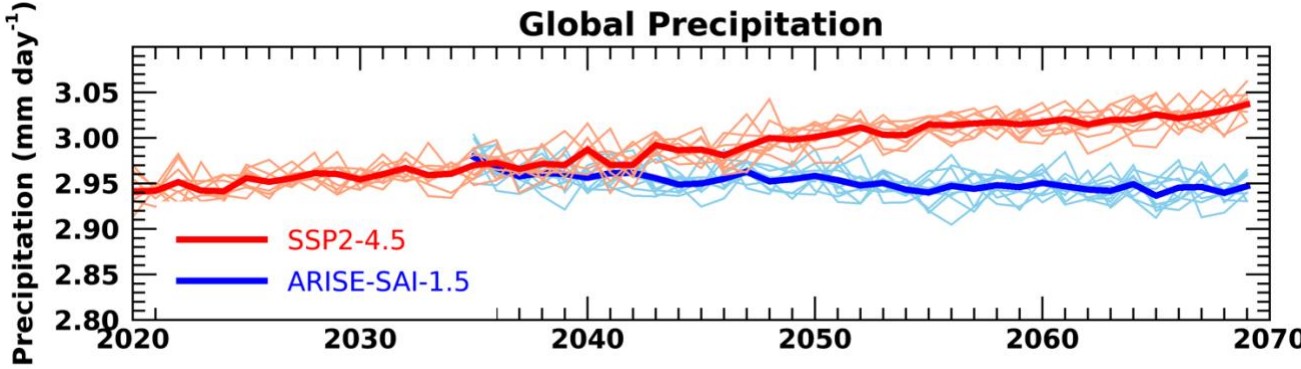

**Figure 5:** Same as Figure 3a but for precipitation.

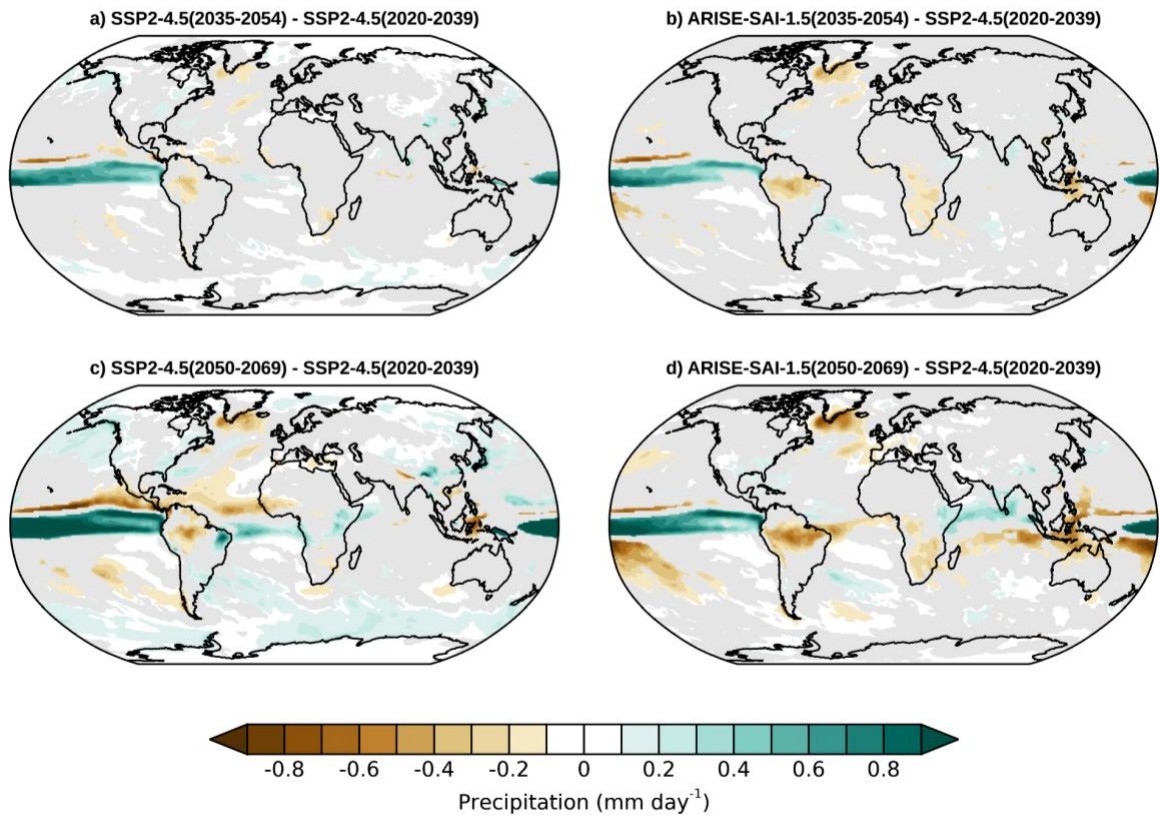

**Figure 6:** Same as Figure 4 but for annual mean precipitation.

**5 Conclusions**

We have described here a detailed new modeling protocol and the first set of simulations of Assessing Responses and Impacts of Solar climate intervention on the Earth system with Stratospheric Aerosol Injection (ARISE-SAI), for studies of impacts of climate intervention using stratospheric aerosols. We have carried out the ARISE-SAI-1.5 simulations utilizing CESM2(WACCM6) and provided extensive output for community analysis. The protocol for simulations described here can be easily implemented in other Earth system models with similar capabilities; furthermore, the protocol can easily be adapted to explore different climate intervention scenarios considering other climate targets, such as different global mean cooling targets, and in the future extended to other types of climate intervention, such as marine cloud brightening. The SAI injection strategy defined by the protocol builds on the approach used in GLENS that was carried out with CESM1(WACCM), but uses a more moderate background emissions scenario, a start date of 2035 rather than 2020, and a target temperature of 1.5°C over pre-industrial following the AR6 definition; the set of simulations presented here also uses a newer version of CESM, which is the same as used for CMIP6 (Gettelman et al., 2019). In these new simulations, the $SO_2$ injections required to keep the global mean temperature, interhemispheric temperature gradient, and pole-to-pole temperature gradient at the target level in ARISE-SAI-1.5 are needed primarily at 15°S, in contrast to GLENS which utilized $SO_2$ injections primarily

at 30ºN and 30ºS. The reasons for these differences are currently being investigated in detail, and it highlights the
need to reproduce such experiments with other climate models to understand their sources. Surface climate in ARISE-
SAI-1.5 is very similar to that during the reference period (2020 - 2039), however residual changes still remain, in
particular in the North Atlantic, where surface temperature is cooler than in the reference period. The robustness of
these projected regional residuals in other climate models, or under different climate targets, would also be of extreme
interest. Consistent with prior studies, global mean precipitation in ARISE-SAI-1.5 is smaller than during the reference
period.
The output for the ARISE-SAI-1.5 simulations is extensive and includes variables from multiple Earth system
components enabling the community analysis of changes in many variables that are crucial to making decisions about
the implementation of SCI including weather and climate extremes, crops, ozone changes, etc. To enable broad access
to the data, output from the ARISE-SAI-1.5 simulations is available on the Amazon Web Services Open Data portal.
**Appendix A**

| Variable Name | Description |
|---|---|
| AEROD_v | Total Aerosol Optical Depth in visible band |
| AODVIS | Aerosol optical depth 550 nm, day only |
| BURDENSO4dn | Sulfate aerosol burden, day night |
| CLDHGH | Vertically-integrated high cloud |
| CLDLOW | Vertically-integrated low cloud |
| CLDMED | Vertically-integrated mid-level cloud |
| CLDTOT | Vertically-integrated total cloud |
| CLOUD | Cloud fraction |
| dgnumwet1 | Aerosol mode (accumulation) wet diameter |
| dgnumwet2 | Aerosol mode (Aitken) wet diameter |
| dgnumwet3 | Aerosol mode (coarse) wet diameter |
| DTCOND | T tendency - moist processes |
| FLDS | Downwelling longwave flux at surface |
| FLDSC | Clearsky Downwelling longwave flux at surface |
| FLNR | Net longwave flux at tropopause |

| | |
|---|---|
| FLNS | Net longwave flux at surface |
| FLNSC | Clearsky net longwave flux at surface |
| FLNT | Net longwave flux at top of model |
| FLNTC | Clearsky net longwave flux at top of model |
| FLUT | Upwelling longwave flux at top of model |
| FLUTC | Clearsky upwelling longwave flux at top of model |
| FSDS | Downwelling solar flux at surface |
| FSDSC | Clearsky downwelling solar flux at surface |
| FSNR | Net solar flux at tropopause |
| FSNS | Net solar flux at surface |
| FSNSC | Clearsky net solar flux at surface |
| FSNTOA | Net solar flux at top of atmosphere |
| FSNTOAC | Clearsky net solar flux at top of atmosphere |
| FSNT | Net solar flux at top of model |
| FSNTC | Clearsky net solar flux at top of model |
| LWCF | Longwave cloud forcing |
| H2O | Water vapor concentration |
| ICEFRAC | Fraction of sfc area covered by sea-ice |
| num_a1 | Aerosol mode (accumulation) number concentration |
| num_a2 | Aerosol mode (Aitken) number concentration |
| num_a3 | Aerosol mode (coarse) number concentration |
| O3 | Ozone concentration |
| O3_Loss | Ozone reaction rate group |
| O3_Prod | Ozone reaction rate group |
| MSKtem | Transformed Eulerian Mean diagnostics mask |
| OMEGA | Vertical velocity (pressure) |
| PBLH | PBL height |
| PHIS | Surface geopotential |

| | |
|---|---|
| PRECC | Convective precipitation rate |
| PRECT | Total (convective and large-scale) precipitation rate |
| PRECTMX | Maximum (convective and large-scale) precipitation rate |
| PS | Surface pressure |
| PSL | Sea level pressure |
| Q | Specific humidity |
| QRL | Longwave heating rate |
| QRL_TOT | Merged LW heating: QRL+QRLNLTE |
| QRS | Solar heating rate |
| QRS_TOT | Merged SW heating: |
| QSNOW | Diagnostic grid-mean snow mixing ratio |
| RELHUM | Relative humidity |
| REFF_AERO | Aerosol effective radius |
| RHREFHT | Reference height relative humidity |
| SO2 | Sulfur dioxide concentration |
| so4_a1 | so4_a1 (accumulation)  concentration |
| so4_a2 | so4_a2 (Aitken) concentration |
| so4_a3 | so4_a3 (coarse) concentration |
| SST | sea surface temperature |
| SWCF | Shortwave cloud forcing |
| T | Temperature |
| TREFHT | Reference height temperature |
| TREFHTMN** | Minimum reference height temperature |
| TREFHTMX** | Maximum reference height temperature |
| TS | Surface temperature (radiative) |
| TROP_P | Tropopause Pressure |
| TROP_T | Tropopause Temperature |
| TSMN | Minimum surface temperature |

| | |
|---|---|
| TSMX | Minimum surface temperature |
| U | Zonal wind |
| U10 | 10m wind speed |
| V | Meridional wind |
| Z3 | Geopotential Height (above sea level) |
| Z500 | Geopotential height at 500 hPa pressure surface |


**Table A1:** Minimum recommended monthly mean output for ARISE-SAI simulations and corresponding reference
simulations.

| Variable Name | Description |
|---|---|
| ACTNL | Average Cloud Top droplet number |
| ACTREL | Average Cloud Top droplet effective radius |
| bc_a4_SRF* | Black carbon in additional mode in bottom layer |
| BURDENBCdn | Black carbon aerosol burden, day night |
| BURDENDUSTdn | Dust aerosol burden, day night |
| BURDENPOMdn | Particulate organic matter aerosol burden, day night |
| BURDENSEASALTdn | Seasalt aerosol burden, day night |
| BURDENSO4dn | Sulfate aerosol burden, day night |
| BURDENSOAdn | SOA aerosol burden, day night |
| BUTGWSPEC | Zonal wind tendency from convective gravity waves |
| CDNUMC | Vertically-integrated droplet concentration |
| CLDICE | Grid box averaged cloud ice amount |
| CLDLIQ | Grid box averaged cloud liquid amount |
| CLDTOT | Vertically-integrated total cloud |
| CLOUD | Cloud fraction |
| CMFMC | Moist convection (deep+shallow) mass flux |
| CMFMCDZM | Convection mass flux from ZM deep |
| dst_a1* | Dust concentration in accumulation mode |

| | |
|---|---|
| dst_a2* | Dust concentration in Aitken mode |
| dst_a3* | Dust concentration in coarse mode |
| dst_a2_SRF* | Aitken mode dust in bottom layer |
| FCTL | Fractional occurrence of cloud top liquid |
| FLDS | Downwelling longwave flux at surface |
| FLDSC | Clearsky Downwelling longwave flux at surface |
| FLNR | Net longwave flux at tropopause |
| FLNS | Net longwave flux at surface |
| FLNSC | Clearsky net longwave flux at surface |
| FLNT | Net longwave flux at top of model |
| FLNTC | Clearsky net longwave flux at top of model |
| FLUT | Upwelling longwave flux at top of model |
| FLUTC | Clearsky upwelling longwave flux at top of model |
| FSDS | Downwelling solar flux at surface |
| FSDSC | Clearsky downwelling solar flux at surface |
| FSNR | Net solar flux at tropopause |
| FSNS | Net solar flux at surface |
| FSNSC | Clearsky net solar flux at surface |
| FSNTOA | Net solar flux at top of atmosphere |
| FSNTOAC | Clearsky net solar flux at top of atmosphere |
| LHFLX | Surface latent heat flux |
| MASS | mass of grid box |
| O3 | Ozone |
| MSKtem | Transformed Eulerian Mean diagnostics mask |
| OMEGA | Vertical velocity (pressure) |
| OMEGA500 | Vertical velocity at 500 hPa |
| PBLH | Planetary boundary layer height |
| PDELDRY | Dry pressure difference between levels |

| | |
|---|---|
| PHIS | Surface geopotential |
| PM25_SRF | PM2.5 in the bottom layer |
| pom_a4_SRF* | Particulate organic matter in additional mode in bottom layer |
| PRECC | Convective precipitation rate |
| **PRECT** | Total (convective and large-scale) precipitation rate |
| PRECTMX | Maximum (convective and large-scale) precipitation rate |
| PS | Surface pressure |
| PSL | Sea level pressure |
| Q | Specific humidity |
| QREFHT | Reference height humidity |
| QSNOW | Diagnostic grid-mean snow mixing ratio |
| RELHUM | Relative humidity |
| **RHREFHT** | Reference height relative humidity |
| SFso4_a1* | surface flux of $SO_4$ in accumulation mode |
| SFso4_a2* | surface flux of $SO_4$ in Aitken mode |
| SFbc_a4* | Surface flux of black carbon in additional mode |
| SFpom_a4* | Particulate organic matter in additional mode |
| SFdst_a1* | Surface flux of dust in accumulation mode |
| SFdst_a2* | Surface flux of dust in Aitken mode |
| SFdst_a3* | Surface flux of dust in coarse mode |
| SHFLX | Surface sensible heat flux |
| SO2 | Sulfur dioxide concentration |
| SOLIN | Solar insolation |
| SOLLD | Solar downward near infrared diffuse to surface |
| SOLSD | Solar downward visible diffuse to surface |
| T | Temperature |
| T500, T700, T850 | Temperature at 500, 700 and 850 hPa respectively |
| TAUBLJX | Zonal integrated drag from Beljaars SGO |

| | |
|---|---|
| TAUBLJY | Meridional integrated drag from Beljaars SGO |
| TAUGWX | Zonal gravity wave surface stress |
| TAUGWY | Meridional gravity wave surface stress |
| TAUX | Zonal surface stress |
| TAUY | Meridional surface stress |
| TGCLDIWP | Total grid-box cloud ice water path |
| THzm | Zonal-Mean potential temperature defined on ilevels |
| TGCLDLWP | Total grid-box cloud liquid water path |
| TMQ | Total (vertically integrated) precipitable water |
| TREFHT | Reference height temperature |
| **TREFHTMN**** | Minimum reference height temperature |
| **TREFHTMX**** | Maximum reference height temperature |
| TS | Surface temperature (radiative) |
| TSMN | Minimum surface temperature |
| TSMX | Minimum surface temperature |
| U | Zonal wind |
| U10 | 10m wind speed |
| UTGWORO | U tendency - orographic gravity wave drag |
| UTGWSPEC | U tendency - non-orographic gravity wave drag |
| UVzm | Meridional flux of zonal momentum: 3D zonal mean |
| UWzm | Vertical flux of zonal momentum: 3D zonal mean |
| Uzm | Zonal mean zonal wind defined on ilevels |
| V | Meridional wind |
| VTHzm | Meridional Heat Flux: 3D zonal mean |
| Vzm | Zonal mean meridional wind defined on ilevels |
| Wzm | Zonal mean vertical wind defined on ilevels |
| Z3 | Geopotential Height (above sea level) |
| Z500 | Geopotential height at 500 hPa pressure surface |


**Table A2:** Available daily averaged output from the atmospheric model in ARISE-SAI-1.5 simulations and SSP2-4.5
CESM2(WACCM6) simulations. Variables marked with a '*' are not available from the first five members of
CESM2(WACCM6) SSP2-4.5 simulations. **indicates variables that are available (but erroneous) in the first five
members of CESM2(WACCM6) SSP2-4.5 simulations. Variables in bold are used to calculate extremes indices such
as those presented in Tye et al. (2022).

| Name of Variable(s) | Variable Description |
|---|---|
| CAPE | Convective available potential energy |
| CIN | Convective inhibition |
| CLDLOW | Vertically-integrated low cloud |
| FLUT | Upwelling longwave flux at top of model |
| PRECT | Total (convective and large-scale) precipitation rate |
| PRECC | Convective precipitation rate |
| PRECSC | Convective snow rate (water equivalent) |
| PRECSL | Large-scale snow rate (water equivalent) |
| PSL | Sea level pressure |
| Q200, Q500, Q700, Q850, Q925 | Specific humidity at 200, 500, 700, 850 and 925 hPa respectively |
| T200, T300, T500, T700, T850, T925 | Temperature at 200, 300, 500, 700, 850 and 925 hPa respectively |
| TMQ | Total (vertically integrated) precipitable water |
| U200, U300, U500, U700, U850, U925 | Zonal wind at 200, 300, 500, 700, 850 and 925 hPa respectively |
| V200, V300, V500, V700, V850, V925 | Meridional wind at 200, 300, 500, 700, 850 and 925 hPa respectively |
| Z200, Z500, Z700, Z850, Z925 | Geopotential height at 200, 500, 700, 850 and 925 hPa respectively |


**Table A3:** 3-hourly averaged output from the atmospheric model in ARISE-SAI-1.5 simulations and additional five
SSP2-4.5 CESM2(WACCM6) simulations. None of the above output is contained in the first five ensemble members
of CESM2(WACCM6) SSP2-4.5 simulations.


| IVT | Integrated water vapor transport |
|---|---|
| PS | Surface Pressure |
| Q* | Specific humidity |
| T* | Temperature |
| TS | Surface temperature (radiative) |
| PSL | Sea level pressure |
| RELHUM* | Relative humidity |
| TMQ | Total (vertically integrated) precipitable water |
| U* | Zonal wind |
| U10 | 10m wind speed |
| uIVT | Zonal water vapor transport |
| vIVT | Meridional water vapor transport |
| V* | Meridional wind |
| Z3* | Geopotential Height |


**Table A4:** 3-hourly instantaneous output from the atmospheric model in ARISE-SAI-1.5 simulations and additional five SSP2-4.5 CESM2(WACCM6) simulations. For the variables marked with a '*', only the bottom-most 22 levels were retained, hence levels for those variables range from 1000 to 103 hPa. None of the above output is contained in the first five ensemble members of CESM2(WACCM6) SSP2-4.5 simulations.


| Name of Variable | Variable Description |
|---|---|
| NO2_SRF | NO2 in bottom layer |
| O3_SRF | O3 in bottom layer |
| PM25_SRF | PM2.5 at the surface |
| PRECC | Convective precipitation rate |
| PRECT | Total (convective and large-scale) precipitation rate |
| TS | Surface temperature (radiative) |


**Table A5:** 1-hourly instantaneous output from the atmospheric model in ARISE-SAI-1.5 simulations and additional
five SSP2-4.5 CESM2(WACCM6) simulations. None of the above output is contained in the first five ensemble
members of CESM2(WACCM6) SSP2-4.5 simulations.


| Variable Name | Description |
|---|---|
| AR | Autotrophic respiration |
| COL_FIRE_CLOSS | Total column-level fire C loss |
| CPHASE | Crop phenology phase |
| DSTDEP | Total dust deposition |
| DSTFLXT | Total surface dust emission |
| DWT_CONV_CFLUX _PATCH | Patch-level conversion C flux |
| DWT_SLASH_CFLUX | Slash C flux to litter and CWD due to land use |
| DWT_WOOD_PROD UCTC_GAIN_PATCH | Patch-level landcover change-driven addition to wood product pools |
| EFLX_LH_TOT | Total latent heat flux |
| FGR | Heat flux into soil/snow including snow melt and lake / snow light transmission |
| FIRA | Net infrared (longwave) radiation |
| FIRE | Emitted infrared (longwave) radiation |
| FROOTC | Fine root carbon |

| | |
|---|---|
| FSH | Sensible heat not including correction for land use change and rain/snow conversion |
| FSR | Reflected solar radiation |
| GDDHARV | Growing degree days needed to harvest |
| GDDPLANT | Accumulated growing degree days past planting date for crop |
| GPP | Gross primary production |
| GRAINC_TO_FOOD | Grain carbon to food |
| H2OSNO | Snow depth (liquid water) |
| HR | Total heterotrophic respiration |
| HTOP | Canopy top |
| NPP | Net primary production |
| Q2M | 2m specific humidity |
| QDRAI | Sub-surface drainage |
| QDRAI_XS | Saturation excess drainage |
| QIRRIG | Water added through irrigation |
| QOVER | Surface runoff |
| QRUNOFF | Total liquid runoff |
| QSNOMELT | Snow melt rate |
| QSOIL | Ground evaporation |
| QTOPSOIL | Water input to surface |
| QVEGE | Canopy evaporation |
| QVEGT | Canopy transpiration |
| RH2M | 2m relative humidity |
| SLASH_HARVESTC | Slash harvest carbon |
| SNOWDP | Gridcell mean snow height |
| SOILWATER_10CM | Soil liquid water + ice in top 10cm of soil |
| TG | Ground temperature |
| TLAI | Total projected leaf area index |
| TOTSOILLICE | Vertically summed soil ice |

| | |
|---|---|
| TOTSOILLIQ | Vertically summed soil liquid water |
| TREFMNAV | Daily minimum of average 2-m temperature |
| TREFMXAV | Daily maximum of average 2-m temperature |
| TSA | 2m air temperature |
| TSKIN | Skin temperature |
| TSOI_10CM | Soil temperature in top 10cm of soil |
| TV | Vegetation temperature |
| TWS | Total water storage |
| U10 | 10-m wind |
| U10_DUST | 10-m wind for dust model |
| URBAN_HEAT | Urban heating flux |
| WASTEHEAT | Sensible heat flux from heating/cooling sources of urban waste heat |
| WOOD_HARVESTC | Wood harvest carbon |


**Table A6:** Available daily averaged output from the land model at landunit-level in ARISE-SAI-1.5 simulations and
additional five SSP2-4.5 CESM2(WACCM6) simulations. None of the above output is contained in the first five
ensemble members of CESM2(WACCM6) SSP2-4.5 simulations.

| | |
|---|---|
| CPHASE | Crop phenology phase |
| CROPPROD1C | 1-yr grain product carbon |
| CWDC_vr | Coarse woody debris carbon, vertically resolved) |
| CWDN_vr | Coarse woody debris nitrogen (vertically resolved) |
| EFLX_LH_TOT | Total latent heat flux |
| FGR | Heat flux into soil/snow including snow melt and lake / snow light transmission |
| FPSN | Photosynthesis |
| FROOTC | Fine root carbon |
| FSH | Sensible heat not including correction for land use change and rain/snow conversion |
| FSNO_ICE | Fraction of ground covered by snow |

| | |
|---|---|
| GDDHARV | Growing degree days needed to harvest |
| GDDPLANT | Accumulated growing degree days past planting date for crop |
| GPP | Gross primary production |
| GRAINC | Grain carbon |
| H2OSOI | Volumetric soil water |
| HTOP | Canopy top |
| LEAFC | Leaf carbon |
| LEAFN | Leaf Nitrogen |
| LITR1C_vr, LITR2C_vr, LITR3C_vr | Amount of carbon in litter in different decomposition pools, vertically resolved |
| LITR1N_vr, LITR2N_vr, LITR3N_vr | Amount of nitrogen in litter in different decomposition pools, vertically resolved |
| LIVESTEMC | Live stem carbon |
| PCT_CFT | % of each crop on the crop landunit |
| PCT_GLC_MEC | % of each GLC elevation class on the glc_mec landunit |
| PCT_LANDUNIT | % of each landunit on grid cell |
| PCT_NAT_PFT | % of each PFT on the natural vegetation (i.e., soil) landunit |
| QICE_FORC | Surface mass balance of glaciated grid cells forcing sent to the glacier model |
| QIRRIG | Water added through irrigation |
| RAIN | Atmospheric rain, after rain/snow repartitioning based on temperature |
| Rnet | Net radiation |
| SMINN | Soil mineral N |
| SMP | Soil matric potential |
| SOILC_vr | SOIL C (vertically resolved) |
| SOILN_vr | SOIL N (vertically resolved) |
| TLAI | Total projected leaf area index |
| TOPO_FORC | Topographic height sent to glacier model |

| | |
|---|---|
| TOTLITC | Total litter carbon |
| TOTSOMC | Total soil organic matter carbon |
| TOTVEGC | Total vegetation carbon, excluding cpool |
| TOT_WOODPRODC | Total wood product carbon |
| TREFMNAV | Daily minimum of average 2-m temperature |
| TREFMXAV | Daily maximum of average 2-m temperature |
| TSA | 2m air temperature |
| TSAI | Skin temperature |
| TSRF_FORC | Surface temperature sent to glacier model |
| TV | Vegetation temperature |


**Table A7:** Available daily averaged output from the land model at gridcell-level in ARISE-SAI-1.5 simulations and
additional five SSP2-4.5 CESM2(WACCM6) simulations. None of the above output is contained in the first five
ensemble members of CESM2(WACCM6) SSP2-4.5 simulations.


| Name of Variable | Variable Description |
| --- | --- |
| EFLX_LH_TOT | Total latent heat flux |
| FSH | Sensible heat not including correction for land use change and rain/snow conversion |
| H2OSNO | Snow depth (liquid water) |
| H2OSOI | Volumetric soil water |
| QDRAI | Sub-surface drainage |
| QDRAI_XS | Saturation excess drainage |
| QOVER | Surface runoff |
| QRUNOFF | Total liquid runoff |
| QSNOMELT | Snow melt rate |
| QSOIL | Ground evaporation |
| QTOPSOIL | Water input to surface |
| QVEGE | Canopy evaporation |
| QVEGT | Canopy transpiration |
| SOILICE | Soil ice |
| SOILLIQ | Soil liquid water |
| SOILWATER_10CM | Soil liquid water and ice in top 10cm of soil |
| TOTSOILICE | Vertically summed soil cice |
| TOTSOILLIQ | Vertically summed soil liquid water |
| TWS | Total water storage |



**Table A8:** 6-hourly averaged output from the land model in ARISE-SAI-1.5 simulations and additional five SSP2-
4.5 CESM2(WACCM6) simulations. None of the above output is contained in the first five ensemble members of
CESM2(WACCM6) SSP2-4.5 simulations.






| Name of Variable | Variable Description |
|---|---|
| CaCO3_form_zint_2 | Total CaCO3 formation vertical integral |
| diatChl_SURF | Diatom chlorophyll surface value |
| diatC_zint_100m | Diatom carbon 0-100m vertical integral |
| diazChl_SURF | Diazotroph chlorophyll surface value |
| diazC_zint_100m | Diazotroph carbon 0-100m vertical integral |
| DpCO2_2 | Atmosphere-ocean difference in the partial pressure of CO2 |
| ECOSYS_IFRAC_2 | Ice fraction for ecosystem fluxes |
| ECOSYS_XKW_2 | Gas transfer velocity computed based on wind speed squared for ecosys fluxes |
| FG_CO2_2 | Dissolved inorganic carbon surface gas glux |
| photoC_diat_zint_2 | Diatom carbon fixation vertical integral |
| photoC_diaz_zint_2 | Diazotroph carbon fixation vertical integral |
| photoC_sp_zint_2 | Diatom carbon fixation vertical integral |
| spCaCO3_zint_100m | Small Phyto CaCO3 0-100m vertical integral |
| spChl_SURF | Small phyto chlorophyll surface value |
| spC_zint_100m | Small phyto carbon 0-100m vertical integral |
| STF_O2_2 | Dissolved oxygen surface flux |
| zooC_zint_100m | Zooplankton carbon 0-100m vertical integral |
| HMXL_DR_2 | Mixed-Layer depth |
| SSS | Sea surface salinity |
| SST | Surface potential temperature |
| SST2 | Surface potential temperature**2 |
| XMXL_2 | Diazotroph carbon fixation vertical integral |


**Table A9:** Daily averaged output from the ocean model in ARISE-SAI-1.5 simulations and all SSP2-4.5 CESM2(WACCM6) simulations.




| Name of Variable | Variable Description |
| --- | --- |
| aice_d | cce area (aggregate) |
| aicen_d | ice area, categories |
| apond_ai_d | melt pond fraction of grid cell |
| congel_d | congelation ice growth |
| daidtd_d | area tendency dynamics |
| daidtt_d | area tendency thermodynamics |
| dvidtd_d | volume tendency dynamics |
| dvidtt_d | volume tendency thermodynamics |
| frazil_d | frazil ice growth |
| fswabs_d | snow/ice/ocn absorbed solar flux |
| fswdn_d | down solar flux |
| fswthru_d | shortwave through the sea ice to ocean |
| hi_d | grid cell mean ice thickness |
| hs_d | grid cell mean snow thickness |
| ice_present_d | fraction of time-avg interval that ice is present |
| meltb_d | basal ice melt |
| meltl_d | lateral ice melt |
| melts_d | top snow melt |
| meltt_d | top ice melt |
| sisnthick_d | sea ice snow thickness |
| sispeed_d | ice speed |
| sitemptop_d | sea ice surface temperature |
| sithick_d | sea ice thickness |
| siu_d | ice x velocity component |
| siv_d | ice y velocity component |
| vicen_d | ice volume, categories |
| vsnon_d | snow depth on ice, categories |


**Table A10:** Daily averaged output from the sea-ice model in ARISE-SAI-1.5 simulations and all SSP2-4.5
CESM2(WACCM6) simulations.

**Code Availability**

CESM2(WACCM6) is freely available from https://www.cesm.ucar.edu/. CESM tag cesm2.1.4-rc.08 was used to
carry out the simulations and is also available at: https://zenodo.org/record/7271743#.Y2FBIC-B3qA. Python scripts
to generate the case directories with appropriate model tags and output can be found at
https://zenodo.org/record/6474201. The code for the $SO_2$ injections controller can be downloaded from
https://zenodo.org/record/6471092#.Yl76rPPMKQc.


**Data Availability**
All the data presented in this manuscript are available at https://zenodo.org/record/6473954#.YmCAwy-B3qA
from the CESM2(WACCM6) SSP2-4.5 simulations and at https://zenodo.org/record/6473775#.YmCAdy-B3qA
from the ARISE-SAI-1.5 simulations. Complete output from all 10 members of CESM2(WACCM6) SSP2-4.5
simulations and ARISE-SAI-1.5 simulations is freely available the NCAR Climate Data Gateway at
https://doi.org/10.26024/0cs0-ev98 and https://doi.org/10.5065/9kcn-9y79 respectively. The ARISE-SAI-1.5 and
SSP-4.5 datasets are additionally available for free download through the Amazon/AWS Open Data program. These
can be accessed at https://registry.opendata.aws/ncar-cesm2-arise/. We anticipate community analysis of various
aspects of the Earth system of the ARISE-SAI-1.5 simulations. There is no obligation to inform the project authors
about the analysis you are performing, but it would be helpful to reach out to DV in order to coordinate analysis and
avoid duplicate efforts.

**Author contribution**
JR designed and carried out simulations, compiled output requests, created most of the figures, and drafted the
manuscript. DV set-up the injection controller, carried out simulations, created a figure, and wrote parts of the
manuscript. DM co-designed the simulations and helped with interpretation of results. DB created the time series of
and archived all the data. NR created namelists with desired output and scripts to easily set-up the simulations. BD set
up the AWS data hosting site and transferred all the output there. WL analyzed the control simulations and provided
targets for the controller. MT and JL gave input to simulation design and data output requests. All authors reviewed
the                                                                                              manuscript.

**Competing interests**
The authors declare that they have no conflict of interest.

**Acknowledgements**

This material is based upon work supported by the National Center for Atmospheric Research, which is a major facility sponsored by the National Science Foundation under Cooperative Agreement no. 1852977 and by SilverLining through its Safe Climate Research Initiative. The Community Earth System Model (CESM) project is supported primarily by the National Science Foundation. Computing and data storage resources, including the Cheyenne supercomputer (doi:10.5065/D6RX99HX), were provided by the Computational and Information Systems Laboratory (CISL) at NCAR. Cloud storage support is provided through the Amazon Sustainability Data Initiative. We thank two anonymous reviewers for their comments that improved the manuscript (https://doi.org/10.5194/egusphere-2022-125-RC1, https://doi.org/10.5194/egusphere-2022-125-RC2).

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
