# Peer review of "Assessing Responses and Impacts of Solar climate intervention on"

_EGUsphere, 2022_

## Referee Comment (RC1)

**Review of "Assessing Responses and Impacts of Solar climate intervention on the Earth system with stratospheric aerosol injection" (ARISE-SAI) by Richter *et al.**

In this paper, Richter et al. introduce a new stratospheric aerosol injection (SAI) simulation protocol that can readily be adopted in climate models. Specifically, the protocol (ARISE-SAI-1.5) involves maintaining global warming at the Paris target of 1.5°C against the backdrop of a middle of the road emissions scenario (SSP2-4.5) using SAI and an advanced feedback controller. The difference between ARISE-SAI-1.5 and previous SAI model intercomparison projects is that in ARISE-SAI-1.5 the emissions scenario is more consistent with projections than in, e.g., GLENS, and that ARISE-SAI-1.5 uses an advanced feedback controller to reduce residual climate changes, unlike in GeoMIP. Richter et al. then perform simulations with the CESM2(WACCM6) climate model using the SSP2-4.5 and ARISE-SAI-1.5 protocol and 10 ensemble members and provide a preliminary analysis of climate impacts in those simulations.

The ARISE-SAI-1.5 simulation protocol certainly represents a marked improvement over GLENS and GeoMIP such that it is more realistic, and therefore will provide more policy-relevant answers than those previous projects. While the paper contains much detail of interest and is generally well written, I found the structure of the paper to be rather confusing which obscures its overall purpose. To be clear, I think that the paper is very interesting and scientifically valid but needs a simple restructuring, as I outline in the General Comments below, before it is ready to be published.

**General comments**

The paper singularly concentrates on the "ARISE-SAI-1.5" simulation, which should be reflected in the title rather than the umbrella term "ARISE-SAI" as I assume that in future other simulations will be added to the ARISE-SAI project. The title is also rather uninformative and should reflect that the paper both describes a protocol (i.e., ARISE-SAI-1.5) and further includes preliminary analyses – maybe "Protocol and preliminary results from the policy-driven "Assessing Responses and Impacts of Solar climate intervention on the Earth system with Stratospheric Aerosol Injection" (ARISE-SAI) project"? At present, it is unclear what the scope of the paper is from either the title, the abstract, or the introduction.

The abstract is not particularly comprehensive. For instance, it is missing the overall justification for the ARISE-SAI project, something like "previous SAI simulation protocols have been overly simplistic and not particularly policy relevant, instead concentrating on identifying structural differences between climate models" or something to that effect. Additionally, the model used to perform these preliminary analyses (CESM2(WACCM6)) should be mentioned in the abstract, as it is unclear how many models have contributed to this paper.

The end of the introduction is rather weak, and I was left unclear as to what the structure or content of the paper was going to be. Additionally, the overall structure of the paper is rather confusing. Instead, I think the paper should be structure as follows

1. Introduction and justification for new geoengineering project
2. ARISE-SAI project overview and ARISE-SAI-1.5 protocol
3. Model (CESM2(WACCM6) configuration (including location of output)

4. Preliminary climate analyses
5. Conclusions

Currently, the ARISE-SAI-1.5 protocol is buried in section 2, after the model description and so it makes it seem that the CESM2(WACCM6) simulations are the emphasis of the paper, rather than the simulation protocol itself. This may be the reason that Richard Rosen left a comment lamenting the lack of seasonal climate impacts analysis in the paper. To be clear, I think more emphasis should be on the novelty of the new protocol and ARISE project, and that the preliminary climate analyses (including the description of CESM2(WACCM6) should be of secondary importance in the paper.

Related to this, I was disappointed at the lack of discussion over the wider ARISE project, seeing as this is the first paper on something that seems to be a rather comprehensive multinational project. What other simulations are planned for this project? Which other groups are currently contributing to the project and which other climate models? What is the scope or mantra of the project, and will other temperature or climate targets be considered (ARISE-SAI-2.0)? What is the length of the project and are there any time constraints for potential contributors (e.g., initial simulations should be submitted by MM/YY)? Greater clarity here would be very beneficial. Also, more emphasis on the novelty of the project would be helpful to the paper.

Just to be clear, I do not support the previous comment by Richard Rosen saying that the paper should investigate seasonal impacts of SAI on precipitation and temperature, or air quality, etc. That is clearly outside the scope of this paper which I feel should emphasize that the preliminary climate results are of secondary importance to the description of the ARISE-SAI-1.5 protocol. I do not think that the authors need to add anything extra in terms of new climate analyses as that is clearly outside not conducive to the goals of the paper (and there are many precedents for this), but maybe an extra line in the conclusions listing further analyses that could be performed would be beneficial.

Lastly, on a minor note, it would help readability for me if the paragraphs all used "justified" alignment, rather than the mixture of justified and left aligned. This is of course a minor quibble.

**Specific comments**

[L14] "reduce *some of* the consequences of climate change." – 'some of' is rather colloquial. Ideally climate intervention would reduce all of the consequences of climate change, or at least those seen to be detrimental. Consider rephrasing.

[L16] Line beginning "We present here a new modelling protocol and a 10-member…" – this sentence is long and confusing. I recommend that it is broken up and reworded

[L23] "We present here the detailed set-up, aerosol injection strategy, and mean surface climate changes in these simulations so they can be reproduced in other global models.". One thing that is missing from the abstract is justification for running ARISE-SAI-1.5 – i.e., the fact that it is more policy relevant than GLENS and represents an improvement over those simulations. Additionally, the fact that only temperature and precipitation are analysed in this paper should be mentioned here. Lastly, add the name of the model into the abstract. I also recommend mentioning that a feedback algorithm is part of the protocol.

[L30] "Stratospheric aerosol injection (SAI) has been shown to be a promising method of global climate intervention in terms of restoring climate to present day conditions" – but only in

climate models and on top of this only in certain climate models and under specific SRM implementations. You could also mention the volcanic analogue here

[L50] "The Geoengineering Large Ensemble (GLENS, Tilmes et al. 2018)" - I recommend introducing a paragraph break here

[L69] "Here we describe a new set-up of an ensemble of simulations with CESM2(WACCM6) designed to simulate a more plausible implementation scenario of SCI using SAI that can be replicated by other modelling centres, and present preliminary diagnostics to begin enabling community assessment of responses of the Earth system to such an intervention." – the end confuses the purpose of the paper. Firstly, the ARISE-SAI-1.5 protocol should be differentiated from the CESM2(WACCM6) ensemble of simulations, which is a single realisation of the protocol (albeit the first). The following, I think, reads better.

"Here we propose a new SAI simulation protocol (ARISE-SAI-1.5) which can be readily implemented in climate models, and we describe realisations of ARISE-SAI-1.5 in the CESM2(WACCM6) climate model. The paper is structured as follows: …"

[L76] "Model description" – I think the ARISE-SAI-1.5 description should come before the model description (see General Comments)

[L96] "CESM2(WACCM6) includes prognostic aerosols which are represented using the Modal Aerosol Model version 4 (MAM4) as described in Liu et al. (2016)." - The aerosol scheme is fundamental to the paper and should be afforded more description, i.e., how many modes, what species, how detailed is the sulphur cycle, etc.

[L131] "We carried out an additional 5-member ensemble of these simulations from years 2015 – 2069 with augmented high-frequency output for high-impact event analysis, as well as additional output for the land model to match the SCI simulations" – is this part of the ARISE-SAI-1.5 simulation protocol? Do you expect other models to perform the high-frequency simulations? What subset of variables are needed as output from other participating climate models? Please be specific about the ARISE-SAI-1.5 protocol.

[L143] "we denote the entire planned set of simulations as "Assessing Responses and Impacts of Solar climate intervention on the Earth system," or "ARISE," with simulations of SAI denoted "ARISE-SAI". – this would be a good place to suggest what other climate interventions could or will be considered. MAC, CCT, CDR etc? Also, are other SAI simulations being planned?

[L148] I recommend moving "possibly" as 1.5°C is seen as a critical threshold

[L154] "These four injection locations are sufficient to independently control the targets that we are trying to achieve" – this is certainly true in CESM1 but has not been verified in other climate models or indeed in the earth system itself. Please add a caveat.

[L204] "GLENS also required more at 30oN/30oS to maintain T2" – more "SO2 injection"

[L306] "The largest exception to that is the North Atlantic warming hole" - Why does AMOC weaken even further under SAI? Or is it that it weakens the same but is not compensated by global warming?

[L323] Consider combining Figs 5 and 6

[L349] "Consistent with prior studies, global mean precipitation in ARISE-SAI-1.5 is smaller than during the reference period." – how significant is this difference? It looks pretty small

---

## Author Comment (AC1)

Thank you for your comments. Our responses are in blue.

My first concern about this paper is that it does not discuss seasonal impacts on temperature and precipitation as a function of latitude at all.  This must be done.  For example, we know that the annual average temperature impacts of climate change on annual average temperature occur most in the high northern latitudes, in places like Alaska.  Furthermore, in theory, having more CO2 in the atmosphere over such regions clearly has a huge impact in reducing radiative cooling in winter, thus increasing surface temperatures substantially.  Yet, having sulfate particles over Alaska in winter won't have much impact in reducing temperatures since the periods of sunlight are so short.  Furthermore, having most of the sulfate particulates much farther south, as shown in Figure 2, would seem to imply that incoming summer radiation will not be reflected very much in the far north were it is needed to be reflected during the long daytimes of summer to cool the air.  Similar seasonal assymetries are probably important for precipitation impacts of climate change even though these would be much harder to model accurately.  The seasonal assymetries with regard to surface temperature seem to derive much more simply from the physics of CO2 concentrations and the density of sulfate particles in the air.  Thus, concluding that "on average" over the year or over many years solar geoengineering can mitigate climate change is not very helpful when trying to analyze the impact of climate change on human society and the ecology.  Seasonal and time of day (day vs. night) differences in impact on temperature and precipitation are very important to consider.

We completely agree with you that the seasonal impacts on temperature and precipitation must be explored, and they have been in previous simulations (for instance, see Simpson et al., 2019, Visioni et al., 2020). The reason they are not included in this manuscript is that the purpose of this paper is to provide an overview of the simulations and experimental protocol, and we expect subsequent publications will address detailed changes, including seasonal changes. In particular, detailed calculations of extreme temperature and precipitation changes are also in progress (and will be coming in forthcoming manuscripts). Since this topic is clearly important to you, we include the plots of seasonal precipitation and temperature changes in this response for the time period (2050 - 2069) - (2020-2039) in Figures 1 and 2 below. We will wait for the editor's comments on whether they should be included in the revised manuscript.

My second concern is that I do not quickly see any discussion of how the impact of sulfate particles on the reflectivity of solar radiation is modelled at different wavelengths, and at different times of the day.

We will add some discussion of this specific topic. In particular, Earth System models treat solar radiation in discrete spectral bands, and especially in terms of model output only offer information for the overall shortwave and longwave radiation incoming (both direct and diffuse) and outgoing. Previous discussions over changes in ratio of diffuse/direct radiation can be found for instance in Visioni et al. (2021). In the visible part of the spectrum, the impact of the added sulfate burden has been discussed in Kravitz et al. (2012).

Also, I do not see any discussion of the impact of continually falling particles have on air quality, human beings breathing the air, and on ecology and agriculture.

Similarly to the first comment, we fully agree that all of these impacts need to be evaluated, however evaluating all of the above suggested impacts simply can not fit into one manuscript. The idea with this overview paper is to get the data out to the community, fully explain the rationale behind our modeling choices and offer detailed information in order for more models to be able to reproduce our results, and have subsequent manuscripts evaluating impacts on all aspects of the Earth system. To this point, our simulations include extensive output for all model components that can be used to investigate more in depth some of the points raised by the reviewer. Some of the points raised by the reviewer have been discussed in Zarnetske et al. (2021), which we will add in the revised manuscript.

**References:**

Kravitz, B., MacMartin, D. G., and Caldeira, K. (2012), Geoengineering: Whiter skies?, Geophys. Res. Lett., 39, L11801, doi:10.1029/2012GL051652.

Simpson, I. R., Tilmes, S., Richter, J. H., Kravitz, B., MacMartin, D. G., Mills, M. J., et al. (2019). The regional hydroclimate response to stratospheric sulfate geoengineering and the role of stratospheric heating. Journal of Geophysical Research: Atmospheres, 124, 12587– 12616. https://doi.org/10.1029/2019JD031093

Visioni, D., MacMartin, D. G., Kravitz, B., Richter, J. H., Tilmes, S., & Mills, M. J. (2020). Seasonally modulated stratospheric aerosol geoengineering alters the climate outcomes. *Geophysical Research Letters*, 47, e2020GL088337. https://doi.org/10.1029/2020GL088337

Visioni, D., MacMartin, D. G., & Kravitz, B. (2021a). Is turning down the sun a good proxy for stratospheric sulfate geoengineering? Journal of Geophysical Research: Atmospheres, 126, e2020JD033952. https://doi.org/10.1029/2020JD033952

[Figure]

Figure 1: Ensemble and seasonal mean surface (2m) temperature differences between SSP2-4.5 (2050-2069) and SSP2-4.5 (2020-2039) (left columns) and ARISE-SAI-1.5 (2050-2069) and SSP2-4.5 (2020-2039) (Right columns) for four seasons: December, January and February (DJF), March, April, May (MAM), June, July, August (JJA), and September, October, November (SON). Season shown is depicted in the figure titles. Gray shading indicates regions where the differences are not statistically significant at the 95% level using a two-sided Student's t test.

[Figure]

Figure 2: Same as Figure 1 but for precipitation.

---

## Author Comment (AC4)

**Response to Anonymous Reviewer #1:**

We would like to thank the reviewer for very thoughtful comments that have helped to improve the quality of this manuscript.

**General comments**

The paper singularly concentrates on the "ARISE-SAI-1.5" simulation, which should be reflected in the title rather than the umbrella term "ARISE-SAI" as I assume that in future other simulations will be added to the ARISE-SAI project. The title is also rather uninformative and should reflect that the paper both describes a protocol (i.e., ARISE-SAI-1.5) and further includes preliminary analyses – maybe "Protocol and preliminary results from the policy-driven "Assessing Responses and Impacts of Solar climate intervention on the Earth system with Stratospheric Aerosol Injection" (ARISE-SAI) project"? At present, it is unclear what the scope of the paper is from either the title, the abstract, or the introduction.

*Thank you for this suggestion. We have changed the title now to "Assessing Responses and Impacts of Solar climate intervention on the Earth system with stratospheric aerosol injection (ARISE-SAI): protocol and initial results from the first simulations" .*

The end of the introduction is rather weak, and I was left unclear as to what the structure or content of the paper was going to be. Additionally, the overall structure of the paper is rather confusing. Instead, I think the paper should be structure as follows

1. Introduction and justification for new geoengineering project
2. ARISE-SAI project overview and ARISE-SAI-1.5 protocol
3. Model (CESM2(WACCM6) configuration (including location of output)
4. 4. Preliminary climate analyses 5. Conclusions
5. Currently, the ARISE-SAI-1.5 protocol is buried in section 2, after the model description and so it makes it seem that the CESM2(WACCM6) simulations are the emphasis of the paper, rather than the simulation protocol itself. This may be the reason that Richard Rosen left a comment lamenting the lack of seasonal climate impacts analysis in the paper. To be clear, I think more emphasis should be on the novelty of the new protocol and ARISE project, and that the preliminary climate analyses (including the description of CESM2(WACCM6) should be of secondary importance in the paper.
6. Related to this, I was disappointed at the lack of discussion over the wider ARISE project, seeing as this is the first paper on something that seems to be a rather comprehensive multinational project. What other simulations are planned for this project? Which other groups are currently contributing to the project and which other climate models? What is the scope or mantra of the project, and will other temperature or climate targets be considered (ARISE-SAI-2.0)? What is the length of the project and are there any time constraints for potential contributors (e.g., initial simulations should be submitted by MM/YY)? Greater clarity here would be very beneficial. Also, more emphasis on the novelty of the project would be helpful to the paper.

Thank you for your thoughtful suggestions. We have restructured the manuscript to (nearly) what you suggested. There is more emphasis now on the protocol and we have also included a discussion of additional simulations being carried out (ARISE-SAI-1.0 and ARISE-SAI-1.5-2045).

Just to be clear, I do not support the previous comment by Richard Rosen saying that the paper should investigate seasonal impacts of SAI on precipitation and temperature, or air quality, etc. That is clearly outside the scope of this paper which I feel should emphasize that the preliminary climate results are of secondary importance to the description of the ARISE-SAI-1.5 protocol. I do not think that the authors need to add anything extra in terms of new climate analyses as that is clearly outside not conducive to the goals of the paper (and there are many precedents for this), but maybe an extra line in the conclusions listing further analyses that could be performed would be beneficial.

Thank you for supporting us not adding any more analysis to the paper. We have added a sentence in the conclusions about further analyses.

Lastly, on a minor note, it would help readability for me if the paragraphs all used "justified" alignment, rather than the mixture of justified and left aligned. This is of course a minor quibble.

We have now made everything 'justified'.

**Specific comments**

"reduce *some of* the consequences of climate change." – 'some of' is rather colloquial. Ideally climate intervention would reduce all of the consequences of climate change, or at least those seen to be detrimental. Consider rephrasing.

We have changed this now to "reduce the worst consequences" (It's unrealistic to expect climate intervention to counteract all of the changes)

Line beginning "We present here a new modelling protocol and a 10-member..." – this sentence is long and confusing. I recommend that it is broken up and reworded

We have now broken this up into 2 sentences.

"We present here the detailed set-up, aerosol injection strategy, and mean surface climate changes in these simulations so they can be reproduced in other global models.". One thing that is missing from the abstract is justification for running ARISE-SAI-1.5 – i.e., the fact that it is more policy relevant than GLENS and represents an improvement over those simulations. Additionally, the fact that only temperature and precipitation are analysed in this paper should be mentioned here. Lastly, add the name of the model into the abstract. I also recommend mentioning that a feedback algorithm is part of the protocol.

We have implemented these changes - thank you for the suggestion.

"Stratospheric aerosol injection (SAI) has been shown to be a promising method of global climate intervention in terms of restoring climate to present day conditions" – but only inclimate models and on top of this only in certain climate models and under specific SRM implementations. You could also mention the volcanic analogue here

We have now added the volcano analog and also 'in climate or Earth system models' to this sentence.

"The Geoengineering Large Ensemble (GLENS, Tilmes et al. 2018)" - I recommend introducing a paragraph break here

Done.

"Here we describe a new set-up of an ensemble of simulations with CESM2(WACCM6) designed to simulate a more plausible implementation scenario of SCI using SAI that can be replicated by other modelling centres, and present preliminary diagnostics to begin enabling community assessment of responses of the Earth system to such an intervention." – the end confuses the purpose of the paper. Firstly, the ARISE-SAI-1.5 protocol should be differentiated from the CESM2(WACCM6) ensemble of simulations, which is a single realisation of the protocol (albeit the first). The following, I think, reads better. "Here we propose a new SAI simulation protocol (ARISE-SAI-1.5) which can be readily implemented in climate models, and we describe realisations of ARISE-SAI-1.5 in the CESM2(WACCM6) climate model. The paper is structured as follows: ..."

We have taken your suggestions and now those sentences read: "Here we propose a new SAI modeling protocol for a suite of simulations designed to simulate a more plausible implementation scenario of SCI using SAI that can be replicated by other modeling centers. We denote the entire planned set of simulations as "Assessing Responses and Impacts of Solar climate intervention on the Earth system," or "ARISE," with simulations of SAI denoted "ARISE-SAI". We anticipate that in the future similar simulations utilizing other climate intervention methods such as Marine Cloud Brightening (MCB) or Carbon Dioxide Removal (CDR), will result in ARISE-MCB or ARISE-CDR simulations, however the modeling of those phenomena using global models are not as advanced as with SAI, hence this may take several years. In addition, we present preliminary results from the first set of these simulations carried out with the Community Earth System Model, version 2 with the Whole Atmosphere Community Climate Model version 6 as its atmospheric component (CESM2(WACCM6)). The paper is structured as follows: section 2 provides an overview of ARISE-SAI protocol including ARISE-SAI-1.5, section 3 describes the model used to describe the realization of ARISE-SAI-1.5 with CESM2(WACCM6), section 4 shows temperature and precipitation in these simulations, and section 5 offers a summary and conclusions."

"Model description" – I think the ARISE-SAI-1.5 description should come before the model description (see General Comments)

Agreed. We have made structural changes per your recommendation.

"CESM2(WACCM6) includes prognostic aerosols which are represented using the Modal Aerosol Model version 4 (MAM4) as described in Liu et al. (2016)." - The aerosol scheme is fundamental to the paper and should be afforded more description, i.e., how many modes, what species, how detailed is the sulphur cycle, etc.

We have added some more information about the aerosol scheme.

"CESM2(WACCM6) includes prognostic aerosols which are represented using the Modal Aerosol Model version 4 (MAM4) as described in Liu et al. (2016). This includes four modes, of which only three are used for sulfate: Aitken, Accumulation and Coarse mode. In the stratosphere, CESM(WACCM6) includes a comprehensive interactive sulfur cycle, as described for instance in Mills et al. (2016); this allows for SO2 oxidation (with interactive OH concentration) and subsequent nucleation and coagulation of H2SO4 into sulfate aerosol (allowing for inter-mode transfer), which are then removed from the stratosphere through gravitational settling and large scale circulation. A more in depth analysis of the size distribution and vertical distribution of sulfate aerosols under SO2 injections has been performed in Visioni et al. (2022) (for single-point injections at the same latitudes and altitudes as those described in these simulations), also compared with results from other models with similar aerosol microphysics (UKESM1 and GISS), highlighting that, in CESM2(WACCM6) the produced stratospheric aerosol are mainly found in the Coarse mode."

"We carried out an additional 5-member ensemble of these simulations from years 2015 – 2069 with augmented high-frequency output for high-impact event analysis, as well as additional output for the land model to match the SCI simulations" – is this part of the ARISE- SAI-1.5 simulation protocol? Do you expect other models to perform the high-frequency simulations? What subset of variables are needed as output from other participating climate models? Please be specific about the ARISE-SAI-1.5 protocol.

We have clarified this now and defined a set of minimal recommended output for the protocol (and left the high frequency output as optional).

"we denote the entire planned set of simulations as "Assessing Responses and Impacts of Solar climate intervention on the Earth system," or "ARISE," with simulations of SAI denoted "ARISE-SAI". – this would be a good place to suggest what other climate interventions could or will be considered. MAC, CCT, CDR etc? Also, are other SAI simulations being planned?

We mentioned now the possibility of ARISE-MCB and ARISE-CDR (which will take some time to implement as those modeling efforts are not as advanced as SAI). Also, we added a subsection 2.5 describing ongoing additional simulations, ARISE-SAI-1.0 and ARISE-SAI-1.5-2045 (with delayed start to 2045).

I recommend moving "possibly" as 1.5oC is seen as a critical threshold

We have now removed 'possibly'.

"These four injection locations are sufficient to independently control the targets that we are trying to achieve" – this is certainly true in CESM1 but has not been verified in other climate models or indeed in the earth system itself. Please add a caveat.

We added the following now:
"These four injection locations have also been demonstrated to be sufficient to produce the optical depth patterns that independently control the targets that we are trying to achieve in various versions of CESM(WACCM) (MacMartin et al., 2017; Zhang et al., 2022; Visioni et al., 2022)."

"GLENS also required more at 30oN/30oS to maintain T2" – more "SO2 injection"

We have corrected this.

"The largest exception to that is the North Atlantic warming hole" - Why does AMOC weaken even further under SAI? Or is it that it weakens the same but is not compensated by global warming?

AMOC is stronger under SAI than it would have been under SSP2-4.5 without SAI, but it does not fully recover back to present-day levels and thus some "warming hole" persists.

Consider combining Figs 5 and 6

Since figure 5 and 6 correspond to Figures 3 and 4 (which are separate), we leave these as they are.

"Consistent with prior studies, global mean precipitation in ARISE-SAI-1.5 is smaller than during the reference period." – how significant is this difference? It looks pretty small

---

## Author Comment (AC5)

**Response to Anonymous Reviewer #2:**

**We thank the reviewer for a positive review!**

The manuscript presents a description of a new experiment to explore the effects of Stratospheric Aerosol Injection (SAI) on climate, part of a larger set of experiments (ARISE) to explore different types of solar radiation management. In particular, the manuscript presents an overview of the experimental setup and a very modest assessment of the results using the CESM2(WACCM6) model. In contrast to a previous SAI experiments (GLENS) run with the CESM1(WACCM) model, the new set of experiments uses a control experiment with a much lower radiative forcing scenario (SSP2-4.5) than the previous set (RCP8.5) thus necessitating a lower level of SO2 injection to stabilize near-surface temperature. Many have argued that the high radiative forcing scenarios like RCP8.5 are unrealistic, and the authors also make the argument that assessing SAI for a moderate radiative forcing scenario may prove to be a more applicable to our future.

The manuscript is very well written and a small selection of results are clearly presented. The few results that are presented, the effects of SAI on a number of measures related to near-surface temperature and precipitation, will be quite familiar to researchers in the field. The most intriguing result, the need for an injection of SO2 much more heavily weighted towards the Southern Hemisphere in this experiment with CESM2(WACCM6)-SSP2-4.5 in comparison with the earlier GLENS experiment with CESM1(WACCM)-RCP8.5 is discussed in a number of places but not illustrated with any results. I understand the reasons for the difference is part of on-going research, but a clearer picture of how the GLENS simulations compares with the ARISE simulations would add considerably to the paper and not compromise any on-going work to understand the reasons for the differences. Figures presenting the same information for the GLENS experiment as is given in Figure 2 (zonal mean stratospheric sulphate distribution) and Figure 3 (time series of T0, T1, and T2) would provide the reader with a much more complete idea of the differences. I would urge the authors to provide a bit more of a description of the differences between the GLENS and ARISE simulations.

You are correct - the difference between SO2 injections between CESM1(WACCM) and CESM2(WACCM6) is very intriguing and this also required us to dig deep into the models & emission scenarios to understand the differences. The reason for differences is not simple to explain, hence we have written a whole separate manuscript strictly focusing on this topic. We couldn't cite it at the time of submission of this paper but we have a citation now to reference: https://egusphere.copernicus.org/preprints/2022/egusphere-2022-779/

This manuscript includes specifically the differences in injections, surface temperature (and other quantities) between GLENS and ARISE and offers some explanation for these differences. We identify three main contributors including: 1) the rapid adjustment of clouds and rainfall to elevated levels of carbon dioxide, 2) the associated low-frequency dynamical responses in the Atlantic Meridional Overturning Circulation, and 3) the contrasts in future climate forcing scenarios. We cite the above manuscript now in the revised manuscript.

The model description in Section 2.1 seems quite detailed for a paper that is ostensibly about a new geoengineering experiment, particularly in describing changes between CESM1 and CESM2. I would suggest this section could be shortened somewhat.

Agreed. We have shortened each section and removed references to CESM1. In response to Reviewer #1 we did add a more detailed description of the aerosol model.

Other than these two suggestions I only have a small number of minor corrections that are listed below.

Minor comments:

Line 68 – I think the reference for Tilmes et al. (2020) is missing from the reference list.

Thank you for catching that - we have added the reference.

Line 127 – Flagging the 'to' in 'trends to not shift'

Thank you - we fixed that.

Lines 163 – 164 – From the text a bit further down it is evident that T0 is used broadly to denote the global mean near-surface air temperature (at line 175) so it should be defined here and not later.

We have corrected this.

---

## Author Response (AR2)

**egusphere-2022-125**   Submitted on 30 Mar 2022
**Assessing Responses and Impacts of Solar climate intervention on the Earth system with stratospheric aerosol injection (ARISE-SAI): protocol and initial results from the first simulations**
Jadwiga Richter, Daniele Visioni, Douglas MacMartin, David Bailey, Nan Rosenbloom, Brian Dobbins, Walker R. Lee, Mari Tye, and Jean-Francois Lamarque

**Response to Editor's Comments:**

Thank you for your careful consideration!

I think that your manuscript is almost ready to be published pending two technical corrections.
- First, please, it would be great if you can store a copy of cesm2.1.4-rc.08 in Zenodo and include the link and DOI in the Code Availability Section.

We have now uploaded tag cesm2.1.4-rc.08 t to Zenodo and provided a link in the Code Availability Section of the manuscript.

- Second, address this comment by the reviewer adding a small modification to the text if needed:
"At lines 226 – 229, the authors state that the temperature targets T0, T1 and T2 are defined based on the reference SSP2-4.5 simulation with WACCM. Does that mean that each group participating in ARISE-SAI-1.5 should define a separate set of temperature targets based on the particular temperature distribution over 2020 – 2039 from their control simulation? My apologies if this is already mentioned in the manuscript, but I do not remember seeing a recommendation."

We added comments in two places to clarify this:

L126 – 131:
We acknowledge that different climate models, with different baseline temperatures and rates of warming, might have different time periods in which they reach 1.5. Nonetheless, we recommend that the best way to achieve a meaningful and easy comparison between different models would be to use always their own model's 2020-2039 SSP2-4.5 period as a baseline over which to calculate the targets their ARISE-SAI-1.5 simulations. This way, the reference period is the same between models and the 2035 start date remains meaningful in every case.

And 234 – 236:
As noted in section 2.3, we recommend that T0, T1, and T2 targets for other models reproducing ARISE-SAI-1.5 simulations are based on the 2020 – 2039 average from their SSP2-4.5 simulations.

We hope that you find the manuscript acceptable now for publication – let us know please if we can answer any more questions.

Thank you!

Jadwiga